# Phase management in single-crystalline vanadium dioxide beams

Run Shi [1,2], Yong Chen[1,2], Xiangbin Cai [2], Qing Lian[1], Zhuoqiong Zhang[1], Nan Shen[1], Abbas Amini[3], Ning Wang [2] & Chun Cheng [1✉]

A systematic study of various metal-insulator transition (MIT) associated phases of $VO_2$, including metallic R phase and insulating phases (T, M1, M2), is required to uncover the physics of MIT and trigger their promising applications. Here, through an oxide inhibitor-assisted stoichiometry engineering, we show that all the insulating phases can be selectively stabilized in single-crystalline $VO_2$ beams at room temperature. The stoichiometry engineering strategy also provides precise spatial control of the phase configurations in as-grown $VO_2$ beams at the submicron-scale, introducing a fresh concept of phase transition route devices. For instance, the combination of different phase transition routes at the two sides of $VO_2$ beams gives birth to a family of single-crystalline $VO_2$ actuators with highly improved performance and functional diversity. This work provides a substantial understanding of the stoichiometry-temperature phase diagram and a stoichiometry engineering strategy for the effective phase management of $VO_2$.

[1] Department of Materials Science and Engineering, Southern University of Science and Technology, Shenzhen, People's Republic of China. [2] Department of Physics and Center for Quantum Materials, Hong Kong University of Science and Technology, Hong Kong, People's Republic of China. [3] Center for Infrastructure Engineering, Western Sydney University, Kingswood, NSW, Australia. ✉email: chengc@sustech.edu.cn

Vanadium dioxide (VO$_2$) is featured with its multiple structural phases as well as diverse phase transition properties with a transition temperature of about room temperature. Compared to well-known metal-insulator transition (MIT) associated insulating phase (monoclinic M1) and metallic phase (rutile R), the other two insulating M2 (monoclinic) and T (triclinic) phases have received little attention, despite their capability to lead to different phase transition behaviors and properties, because of their metastable structures and spatial phase inhomogeneity in film/bulk samples[1–3]. In recent years, controlled domain structures and phase transitions have been achieved in single-crystalline VO$_2$ beams at the single domain level to decouple the effects of external factors. These advancements have assisted to acquire relatively accurate phase diagrams for an in-depth understanding of MIT mechanism and various device applications[4–8]. Therefore, increasing efforts have been made to investigate the intermediate VO$_2$ phases, specifically M2 and T phases, and their interplay in single-crystalline VO$_2$ beam systems. The outcomes can thrive information about the underlying physics of this controversial MIT for extensive device application purposes[9–14]. However, a reliable method to controllably fabricate these VO$_2$ phases is yet lacking which impedes systematic and critical investigations on the phase transitions as well as their high-tech applications.

The stabilization of M2 and T phases can be achieved by doping[15–18], external strain[19,20], and oxygen nonstoichiometry[21,22]. Since either doping or strain causes remarkable structural distortion to VO$_2$ lattices and varies their intrinsic properties and applications, an effective stoichiometric strategy is highly required for engineering the VO$_2$ crystals with stabilized multi-phases. Zhang et al.[21] created an oxygen-rich reaction condition by injecting O$_2$ flow to the vapor transport system during the first 15 min of heating, but only 15% of as-grown VO$_2$ nanowires appeared to have stabilized M2/T phases by excessive oxygen at room temperature, implying the difficulty of the kinetics control of oxidation. Wang et al.[22] and Kim et al.[23] successfully grew M2/T phases in the VO$_2$ beams on specific r-cut sapphire substrates by a two-step vapor transport method; they attributed the formation of these intermediate phases to stoichiometric defects and size effect, respectively. Despite the above studies, the stable and meticulous control of phase structures in VO$_2$ through continuous stoichiometry modulation has yet remained a great challenge.

Here, we report the stoichiometry engineering of single-crystalline VO$_2$ beams through an oxide inhibitor-assisted CVD method, which provides an empirical reaction phase diagram for the controllable fabrication of the structured insulating VO$_2$ phases (M1, T, and M2) stabilized at room temperature. The fabrication of VO$_2$ single crystals with controlled individual or spatially combined phase structures enables the comprehensive investigation and manipulation of their phase transition properties. Referring to the classical stoichiometry–temperature phase diagram, we propose and construct the entire family of single-crystalline VO$_2$ actuators with good performance through the fine phase management of individual VO$_2$ beams. The outcomes indicate the powerful modulation capability of the proposed oxide inhibitor-assisted stoichiometry engineering strategy.

## Results

**Characterizations of VO$_2$ phases.** Figure 1a depicts the structures of four VO$_2$ phases during MIT, where the VO$_2$ lattice is represented by two sets of parallel chains of V$^{4+}$ ions in each phase. In their lattices, every V ion is surrounded by six O ions to form one VO$_6$ octahedron. The high-temperature stable VO$_2$ has a typical R structure with two straight chains. Upon the decrease of temperature, the metallic R phase of VO$_2$ is converted to one of the

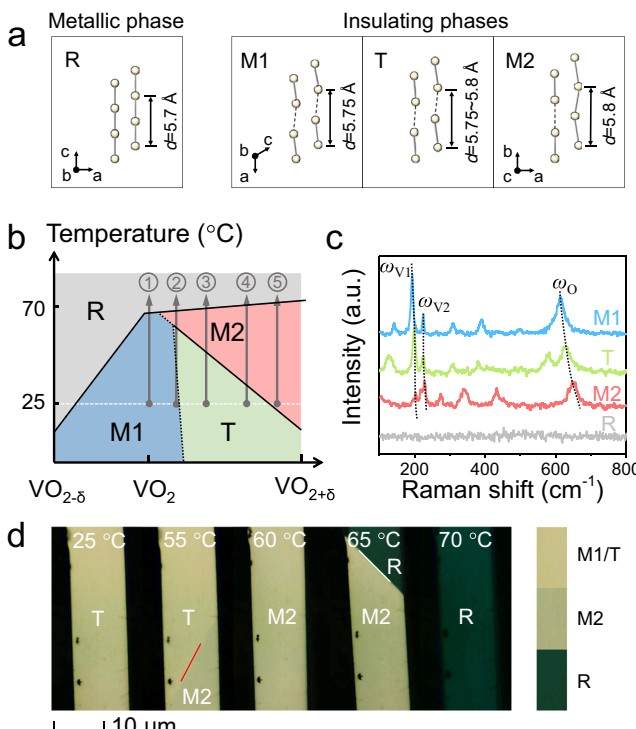

**Fig. 1 Structural configuration and characterization of VO$_2$ phases. a** Schematics of the arrangement of V ions in different VO$_2$ phases (M1, T, M2, and R). The solid circles represent the V ions, solid lines represent a short V–V distance of <0.3 nm, and dash lines connect the neighboring V ions with a V–V distance of >0.3 nm. **b** Schematics of stoichiometry–temperature phase diagram for VO$_2$ crystals. The gray, blue, green, and red regions in the diagram represent the R phase, M1 phase, T phase, and M2 phase, respectively. The gray arrows (routes 1-5) trace the phase evolution process of VO$_2$ with various oxygen contents from room temperature. **c** Characteristic Raman spectra of VO$_2$ phases. Dash lines trace $\omega_{V1}$, $\omega_{V2}$, and $\omega_O$ phonon frequencies in the Raman spectra. **d** Temperature-dependent optical images of a single VO$_2$ beam upon heating, revealing its whole domain evolution process corresponding to route 3 of **b**. The red line shows the position of the domain wall between T and M2 phases, while the white line shows the position of the M2-R domain wall.

three insulating structures, usually the insulating M1 structure via the dimerization of V ions along $c_R$ axis. On the other hand, the M2 structure contains two sublattices of V ions along the chain direction: one of them retains the zigzag V–V chain of the M1 phase, while the straight alignment of V ions in another sublattice is close to the R phase. T phase is an intermediate state between M1 and M2 structures, while its properties are very similar to M1 phase[16]. It has been widely reported that the largest difference between the lattice of these four VO$_2$ phases is along the direction of V–V chain[1,24], i.e., the $c_R$ direction of R phase, with a shared period but different values ($d_R = 5.700$, $d_{M1} = 5.755$, $d_T = 5.755–5.797$ and $d_{M2} = 5.797$, all in Å). These small lattice differences make it hard to distinguish these phases by selected area electron diffraction (SAED) (Supplementary Discussion). It has been reported that the stoichiometry of VO$_2$ significantly affects the phases. Oxygen vacancies can work as electron dopants that stabilize the metallic R phase at low temperature[25], while the high oxygen content and the presence of V$^{5+}$ ions in the VO$_2$ lattice probably lead to the formation of the M2 phase and T phase[21,22]. According to the previous publications[10,21,22], a classical schematic stoichiometry–temperature (S–T) phase diagram is supplied in Fig. 1b as a reference to describe the impact of stoichiometry on VO$_2$ phase structures and the mutual evolution of these phases.

The dashed line shows the opaque transition process at the interface between the M1 and T phases. Five typical phase transition routes are predicted, namely routes 1–5, which will be verified in the latter part to support the validity of the S–T phase diagram.

Vanadium dioxide phases (Fig. 1a) can be well identified by Raman spectroscopy, which is quite sensitive to subtle structural differences. The evidential blue-shift of $\omega_O$ phonon mode from 610 to 650 cm$^{-1}$ is usually used to distinguish the $VO_2$ phases and trace their phase transitions[19] as shown in Fig. 1c. In Raman spectra, the $\omega_O$ frequency of the M1 phase is at <615 cm$^{-1}$ and the M2 phase has the $\omega_O$ mode at >646 cm$^{-1}$, while the range of $\omega_O$ within 615–646 cm$^{-1}$ is attributed to T phase. R phase does not contribute to a detectable Raman signal and thus a flat Raman spectrum indicates its presence. However, Raman measurement cannot achieve a real-time phase identification at a single domain level (in sub-micrometer size) for its long acquisition time (>5 s) and the micrometer-scale laser beam in use. By contrast, the distinct optical contrast of $VO_2$ phases enables the convenient real-time monitoring of the domain evolution process of $VO_2$ crystals during the phase transition via optical microscopy. Figure 1d shows the optical images of the $VO_2$ beam for the T → M2 → R transition route upon heating, corresponding to the predicted route 3 in Fig. 1b. Illuminated by white light, R phase is dark green, M2 phase is dark yellow, and T phase is light yellow, consistent with another report elsewhere[26]. Nevertheless, the optical microscopy method can hardly distinguish T and M1 phases which have very similar colors. Therefore, they should be identified by Raman spectroscopy or the phase transition route according to the S–T phase diagram (Supplementary Discussion). In summary, the optical microscopy, Raman spectroscopy, and the phase transition route characterizations should be applied together to identify phases, and probe the phase transitions and domain dynamics of $VO_2$.

**Stoichiometry engineering of CVD-grown $VO_2$ beams**. In this work, free-standing single-crystalline $VO_2$ beams with selective phase structures are fabricated by a stoichiometry engineering strategy. Wherein, a modified low-pressure CVD reaction is applied and $SiO_2$, as an oxide inhibitor, is uniformly mixed with the V source ($V_2O_5$) to modulate the reaction kinetics (Supplementary Discussion). The $VO_2$ beams prepared by this method have a moderate nucleation density, longer length (hundreds of μm), and larger width (several to tens of μm) than those prepared by conventional CVD methods (Supplementary Fig. 3a). High-resolution transmission electron microscopy (HRTEM), SAED, and energy-dispersive X-ray spectroscopy (EDS) are used to identify the crystal structure and chemical composition of the $VO_2$ products (Supplementary Discussion); they confirm that the as-grown $VO_2$ beams grow along the typical $c_R$ direction.

According to the analysis of the product and precursor composition at different periods of the CVD reaction in Supplementary Discussion, the sketch map is proposed in Fig. 2a for the growth of $VO_2$ beams. This process is performed in two stages: (1) nucleation and growth stage at $T < 850\,°C$, when the nucleation and growth of $VO_2$ are driven by the reduction of high-valence vanadium precursors ($V_2O_5$ or $V_6O_{13}$); (2) stoichiometry modulation stage at $T = 850\,°C$, when the partial pressure of oxygen ($P_{O_2}$) is modulated for the oxidation or deoxidation of the as-grown $VO_2$. Notably, $SiO_2$ can effectively inhibit the evaporation of $V_2O_5$ at stage 2 (Supplementary Discussion), so that most of V sources are locally converted to $VO_2$ within the oxide mixture, while the other product, $O_2$, is released into the reaction atmosphere leading to the increase of $P_{O_2}$. $P_{O_2}$ can be finely modulated during CVD reactions by varying the amount of $SiO_2$: the larger the amount of $SiO_2$, the higher $P_{O_2}$ is. It is noted

that the existence of $V_2O_5$ wetting layer is essential for the oxidation/deoxidation of $VO_2$ beams in stage 2 as discussed in Supplementary Discussion. The resultant uneven distribution of the wetting layer may cause a lateral oxygen gradient throughout $VO_2$ beams (Fig. 2a) with interesting asymmetric domain patterns during the phase transitions[27].

Figure 2b shows that the crystal structure of products is sensitive to the reaction conditions, including reaction time, reaction temperature, and $x$ (the mass ratio of $V_2O_5$ to $SiO_2$). In stage 1, the oxygen-rich condition leads to the formation of room temperature-stable M2 or M2-like T phase ($\omega_O > 635$ cm$^{-1}$). In addition, the decrease of oxygen content in $VO_2$ crystals upon increasing the reaction temperature from 750 to 850 °C, is spotted by the reduced $\omega_O$. In stage 2, the stoichiometry of $VO_2$ crystals varies with different values of $x$. The high value of $x$ (with few $SiO_2$) supports further deoxidation of $VO_2$. This is represented by the reduction of the measured $\omega_O$ frequency to ~612 cm$^{-1}$, similar to the conventional reaction without $SiO_2$[28]. Upon the increase of $SiO_2$ (reduced $x$), $P_{O_2}$ of the reaction system is significantly increased, leading to the formation of oxygen-rich $VO_2$ that is stabilized as M2 phase at room temperature. Notably, this oxide inhibitor-assisted reaction enables a continuous stoichiometry modulation, selectively producing $VO_2$ beams with the desired room temperature-stable phase by adjusting $x$ value at the deposition temperature of 850 °C. Consequently, an empirical phase diagram of $x$ versus deposition time for the fabrication of different room temperature-stable insulating $VO_2$ phases is established in Fig. 2c. In this diagram, the relatively broad region of T phases shows the robust modulation capability of the oxide-assisted stoichiometry engineering method that fundamentally assists the intensive study of this intermediate phase. In conclusion, this stoichiometry modulation strategy enables the phase-selective fabrication of single-crystalline $VO_2$ crystals free of external stress/strain or metal doping; this further benefits the following comprehensive investigation of $VO_2$ phase transition properties.

**Phase transition properties of nonstoichiometric $VO_2$ beams**. The phase evolution processes of typical samples 1–5 obtained by the empirical reaction diagram (Fig. 2c) that contain different oxygen contents are tracked by $\omega_O$ shift as shown in Fig. 2d. They can be well assigned to routes 1–5 in Fig. 1b, and substantially support the validity of the S–T phase diagram. For example, sample 2 follows the route 2 of M1 → T → M2 → R, which is identified by $\omega_O = 610$ cm$^{-1}$ at $T < 35\,°C$, $\omega_O = 610\sim650$ cm$^{-1}$ at 35–50 °C, and $\omega_O = 650$ cm$^{-1}$ at $T > 50\,°C$. Both the M1 phase (sample 1) and M2 phase (sample 5) demonstrate a first-order transition to the R phase, as their characteristic $\omega_O$ peaks suddenly vanish upon heating. By contrast, the phase transitions associated with the T phase show different features: M1 → T transition is a quasi-second-order transition as supported by gradual changes of $\omega_O$ at critical temperature (35 °C for sample 2), where the structural transition between these two phases is quite smooth. T → M2 transition varies from a first-order transition to a gradual transition depending on the initial lattice structure of the T phase. M1-like T phase follows a sharp and fast transition (sample 2), while the M2-like T phase demonstrates a gradual and slow transition (sample 4). In addition, T phase with the frequency $\omega_O$ closer to 650 cm$^{-1}$ has a lower T → M2 transition temperature (50 °C for sample 2, 35 °C for sample 3, and 25 °C for sample 4). It is thus believed that the T phase with a more similar structure to the M2 phase has a smaller energy barrier to overcome in the T → M2 transition. Besides the above difference of phase transition behaviors between the samples with different oxygen contents, it is noted that the structure of the T

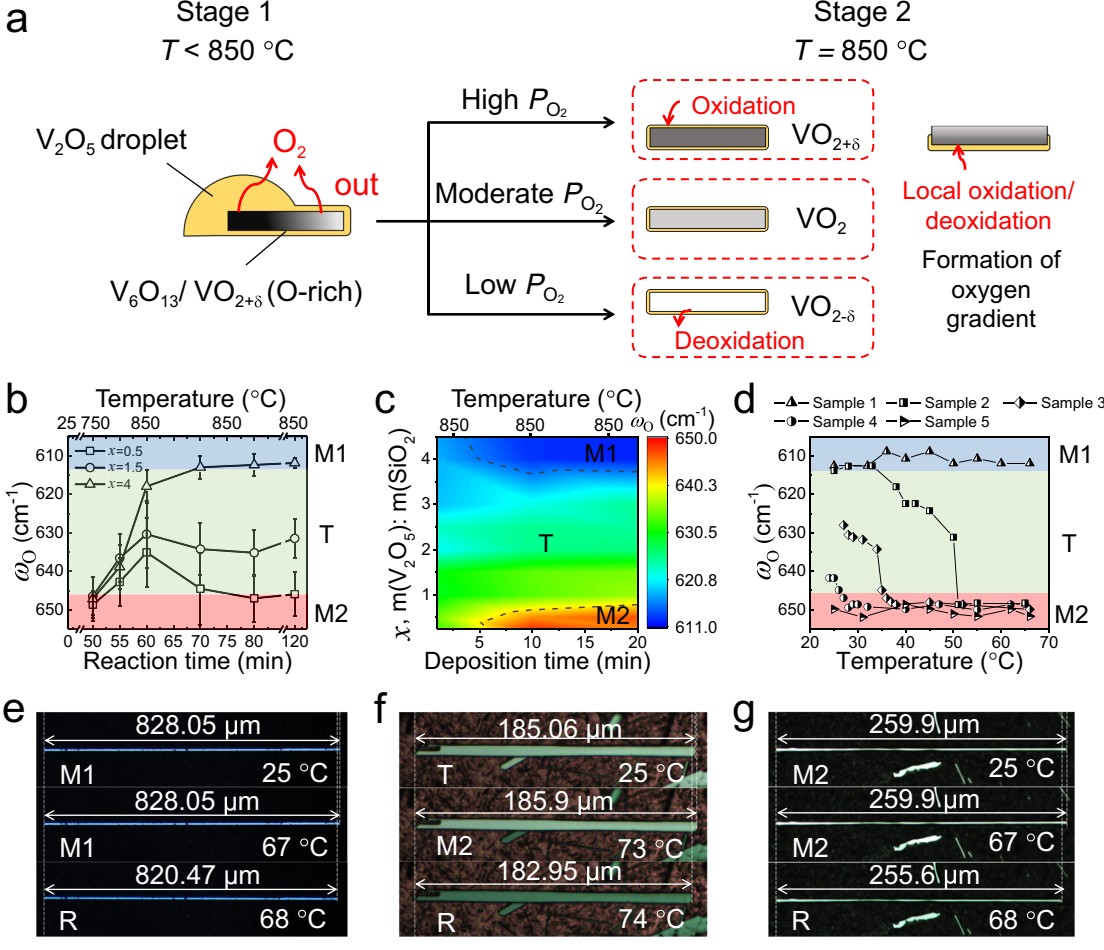

**Fig. 2 Stoichiometry engineering of CVD-grown VO$_2$ crystals. a** Schematics of CVD growth of VO$_2$ beams and their phase transformation under different oxygen partial pressures ($P_{O_2}$s). The reaction process occurs in two stages, stage 1 ($T < 850$ °C) and stage 2 ($T = 850$ °C). The yellow regions represent the V$_2$O$_5$ wetting layer. **b** Phase transformation of as-grown VO$_2$ beams during the typical CVD reaction versus $x$ (the mass ratio of V$_2$O$_5$ to SiO$_2$) and the error bars are the corresponding standard deviations, where the reaction time for the heating process is started at the temperature 25 °C. **c** Structural phase diagram of CVD-grown VO$_2$ beams prepared at 850 °C under different conditions, i.e., $x$ and the deposition time (from the time when the reaction temperature hits 850 °C), which is represented with $\omega_O$ frequency shift. **d** $\omega_O$ frequency shift of the VO$_2$ beams with room temperature-stable M1, T, and M2 phases at increasing temperature, corresponded to routes 1–5 in Fig. 1b. Temperature-dependent optical images of each separated VO$_2$ beam with room temperature-stable **e** M1, **f** T, and **g** M2 phase upon heating.

phase in individual samples gradually changes as the ambient temperature varies (samples 2–4). This fact is revealed by the gradual change of $\omega_O$ (Fig. 2d), consistent with the varied lattice period $d$ (5.755–5.797 Å) of the T phase. Consequently, the apparent stoichiometry modulation in this work is a good tool to investigate the phase transition kinetics associated with the T phase that has been rarely reported. It is also noted that various phases in all samples are finally converted to the R phase at 70 ± 5 °C, suggesting the flat boundary between the R and M2 regions in the S–T diagram of Fig. 1b.

The investigation of fundamental properties such as structure and resistivity change in VO$_2$ beams of these insulating phases upon phase transitions is necessary for their practical applications. The axial strain ($\varepsilon$) of individual VO$_2$ beam across phase transitions can be directly measured by optical microscopy to evaluate the phase uniformity in the entire beam. Figure 2e demonstrates the temperature-dependent optical images of free-standing VO$_2$ beams that are respectively stabilized as M1, T, and M2 phases at room temperature. The magnitude of $\varepsilon$ can be calculated by a simple equation: $\varepsilon = \frac{L_i - L_0}{L_0} \times 100\%$, where $L_0$ is the original length of the beam, and $L_i$ is the final length of beam after the phase transition. It is concluded that the magnitude of $\varepsilon$

for M1 → R transition is $\varepsilon_{M1 \to R} = \sim -0.92\%$ (negative sign indicates the shrinkage), while for M2 → R transition it is $\varepsilon_{M2 \to R} = \sim -1.65\%$. These data are in good agreement with the reported theoretical values (−0.96% for M1 → R and −1.67% for M2 → R)[29]. Given the uncertain crystal structure of T phase, the estimated $\varepsilon_{T \to M2}$ value should be smaller than $\varepsilon_{M1 \to M2}$ (0.73%), which is consistent with the measured $\varepsilon_{T \to M2}$ of ~0.45%. Based on the above results, it is concluded that stoichiometry modulation has been achieved for the entire VO$_2$ beams. Furthermore, the electrical measurements are conducted to evaluate the phase transition properties of the VO$_2$ beams with room temperature-stable M1, T, and M2 phases as shown in Supplementary Fig. 10. All the beams undergo colossal resistivity changes with 4–5 orders of magnitude across MIT, implying the good electrical switching performance of as-grown VO$_2$ beams.

**Single-crystalline VO$_2$ actuators**. The selective phase stability and reliable manipulation of domain kinetics in VO$_2$ crystals greatly extends the applications of MIT in VO$_2$. The single-crystalline VO$_2$ actuator (SCVA) is an ideal example that fully utilizes the phase management; we recently proposed its working principle[27]. In brief, the SCVA is a VO$_2$ beam with a

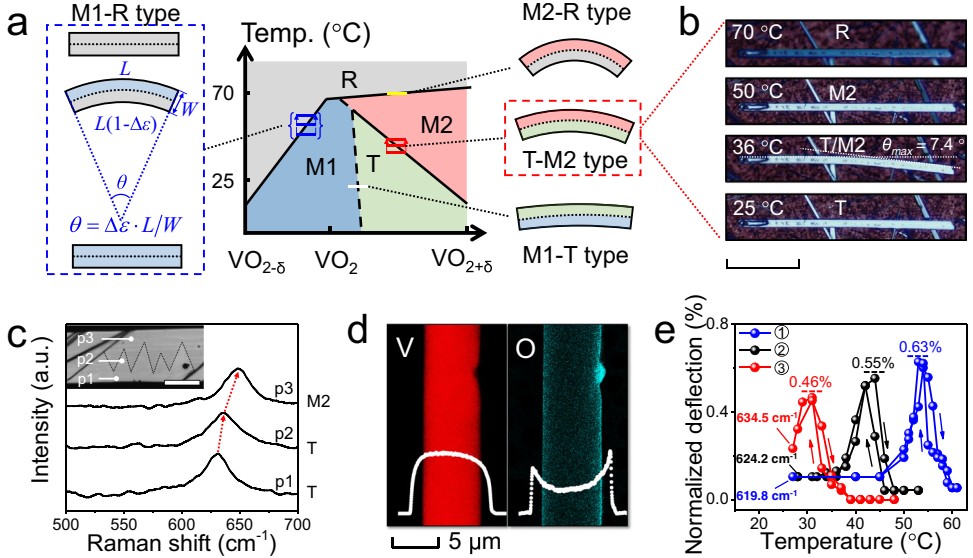

**Fig. 3 Single-crystalline VO₂ actuators (SCVA) driven by asymmetric stoichiometry. a** Working mechanism of SCVAs with asynchronized phase transition routes. The colored bar, namely the gradient bar, in the phase diagram shows the oxygen gradient along the lateral direction of VO₂ beams at a certain temperature. The group of gradient bars at varying temperatures across the phase boundaries represents the effective actuation process of SCVAs. **b** Optical images of T-M2 type SCVA at increasing temperature (scale bar is 100 μm). White dash lines give the maximum deflection angle ($\theta_{max}$) of the actuator at 36 °C. **c** Raman spectra of a bending SCVA. The red dash arrows trace the shift of $\omega_O$ phonon frequency. The inset image shows the measured points (p1–p3) of the sample (scale bar is 10 μm) with black dash-lines at the positions of domain walls between M2 and T domains, where dark yellow domains have the typical M2 structure and shallow yellow domains are T phase. **d** EDS mapping of V and O along a SCVA that bends towards the left side. The white profiles show the element distribution across the lateral direction of beam. **e** Plots of the normalized deflection of three T-M2 SCVAs with different oxygen contents (i.e., room temperature $\omega_O$ frequencies).

stoichiometry gradient along its lateral direction (represented by a gradient bar in the S–T diagram in Fig. 3a), where the two sides of the beam have different phase transition routes. During the heating/cooling, upon the asynchronized phase transitions at the two sides of SCVA, it bends driven by the laterally asymmetric expansion/shrink along the axial direction. Compared to the conventional VO₂ bimorph actuators, SCVAs have a much simpler device structure, predictable actuation, competitive actuation performance, and most importantly, superior stability[27]. The locations of gradient bars in the S–T phase diagram at various temperatures are used to trace the bending process of SCVAs. For example, the array of blue bars in Fig. 3a represents the M1-R type SCVA, which is named upon the coexistence of the M1 and R domains driving the maximum bending state of the beam. The M1-R SCVA demonstrates a straight-bending-straight bidirectional actuation upon heating, corresponding to the M1-M1 → R-M1 → R-R phase evolution process. In an ideal SCVA model with a length of $L$ and width of $W$, the deflection angle ($\theta$) of beam is given by $\theta = \Delta\varepsilon \cdot L/W$ (Fig. 3a), where $\Delta\varepsilon$ is the axial strain difference between the two sides of VO₂ beam. Therefore, the maximum deflection angle ($\theta_{max}$) of a M1-R type SCVA is achieved when the two sides of VO₂ beam are respectively occupied by M1 and R domains, i.e., the $\Delta\varepsilon_{max}$ of M1-R SCVA is equal to $|\varepsilon_{M1\rightarrow R}|$. For comparison, the normalized deflection, defined as the ratio of the deflection angle of a beam to its aspect ratio, is used to describe the thermal-driven actuation of the SCVA and evaluate its performance by comparing the maximum normalized deflection (i.e., $\Delta\varepsilon_{max}$) with the axial strain of the corresponding phase transition. In addition to the bending amplitude, the actuation performance of thermal-driven SCVA should be also analyzed in terms of output work density, actuation speed, energy conversion efficiency, and lifetime.

According to the S–T phase diagram, the entire family of SCVA can be categorized into four types, M1-R, M1-T, T-M2 and M2-R, based on the possible phase configuration at the two sides

of the beams (Fig. 3a). In the S–T phase diagram, if the gradient bar crosses over a phase boundary, the corresponding VO₂ beam can potentially work as a SCVA. Here, owing to the high fabrication temperature and the narrow width of the beam, only small stoichiometry gradients can be practically formed. Therefore, not all types of SCVAs can work as expected with an effective bending. The M1-T SCVA (represented by a white bar) hardly presents evident bending because of the small lattice difference between the two phases. In addition, the colossal strain of ~1.67% of M2-R transition indicates the best performance of the M2-R SCVA among the four types. However, the flat M2-R phase boundary makes it hard to achieve the asynchronized transition routes at the two sides of VO₂ beams. As reported[27], among samples with a small gradient bar, only M1-R SCVA and T-M2 SCVA (red bars in Fig. 3a) are practically applicable.

It is discovered that 20–60% of the VO₂ beams prepared by the inhibitor-assisted CVD method can work as T-M2 SCVAs as shown in Fig. 3b and Supplementary Movie 1. This observation demonstrates their straight-bending-straight bidirectional actuation capability upon heating as predicted in Fig. 3a. Notably, the M2 → R transition of this beam is still a typical first-order transition without any bending, verifying the above assumption that the flat M2-R boundary cannot contribute to the self-bending of SCVAs. With the $L/W$ ~29 and $\theta_{max}$ ~ 7.4° (0.13 rad) of the beam, $\Delta\varepsilon_{max}$ is calculated as ~0.44% that is almost the same as the measured strain of ~0.4% across T → M2 transition. This result suggests that T and M2 domains simultaneously occupy one side of the beam, to achieve an optimal bending actuation. This conclusion is verified by the magnified optical image and Raman spectra of the bending SCVA in Fig. 3c. The EDS element mapping results in Fig. 3d show that the left part of the VO₂ beam has a higher V intensity and lower O intensity than the right part; this fact verifies the oxygen gradient along the lateral direction of the beam. In addition, the strong oxygen intensity at the two sides should be attributed to the V₂O₅ layer on the surface of the beam

(Supplementary Fig. 7a). The suggested growth mechanism of oxygen gradient $VO_2$ beams is discussed in Supplementary Discussion. Figure 3e shows the temperature-dependent actuation of three T-M2 SCVAs which have different oxygen contents and demonstrate dissimilar actuation behaviors. It is noted that with the increase of oxygen content, both the maximum normalized deflection and working temperature of SCVAs decrease; this is due to the reduced lattice difference between T and M2 phases. All in all, it is easy to control the performance of T-M2 SCVAs using the presented stoichiometry engineering strategy.

To utilize the large strain of ~1.67% across M2-R transition and acquire the performance limit of $VO_2$-based actuators[30], it is required to have M2 and R phases coexisted within a single $VO_2$ beam. There are two possible approaches: (1) extending the width of gradient bar with only crossing over the M2-R boundary (yellow dash bar); (2) extending the width of gradient bar so that it crosses over both M1-R and M2-R boundaries (yellow solid bars) as shown in Fig. 4a. The enlarged stoichiometry gradient (wide gradient bar) was achieved by adding an appropriate amount of $WO_2$ to the inhibitor-assisted reaction system (see details in "Methods" section), where the M2-R SCVA was successfully fabricated as shown in Fig. 4b and Supplementary Movie 2. It is discovered that 40-95% of the as-prepared $VO_2$ beams can work as M2-R SCVAs, indicating that W-doping favors the formation of stoichiometry gradient in $VO_2$ beams. By the way, W-doped $VO_2$ should have different stoichiometry–temperature phase diagrams from the applied one (without W-doping), so the yellow gradient bars may not completely depict the transition process of as-grown W-doping $VO_2$ beams.

The as-prepared M2-R SCVA demonstrates a similar bidirectional actuation mode of other kinds of SCVAs, with a clear laterally asymmetric domain configuration across the MIT (Fig. 4b). The temperature-dependent optical images in Fig. 4c and Supplementary Movie 3 show the domain evolution process of SCVA. At the beginning of the heating stage, the formation of a radially asymmetric M2-T-M1 domain pattern contributes to the initial bending (referring to the Raman spectra in Supplementary Fig. 13). With further increase in temperature, R domains gradually occupy the oxygen-deficient side while the oxygen-rich side is occupied by M2 domains, reaching a maximum normalized deflection of ~1.66% close to the theoretical strain of M2-R transition (Fig. 4d). As the temperature hits 60 °C, the entire $VO_2$ beam is taken up by the pure R phase and reforms back to the straight state. This domain evolution process is almost consistent with the yellow solid bars (approach 1) in Fig. 4a, which confirms that the oxygen gradient is expanded by the addition of $WO_2$. However, based on approach 2, the M2-R SCVA is not structured, which may be attributed to relatively flat M2-R phase boundary or distorted S–T diagram upon W-doping. Taking the advantage of the colossal M2-R strain, the as-grown M2-R SCVA demonstrates better actuation performance than the T-M2 and M1-R SCVAs (Fig. 4e). For example, T-M2 and M1-R SCVAs with an aspect ratio of ~56 have $\theta_{max}$ of 13° (0.23 rad) and 24° (0.42 rad), respectively, while M2-R SCVA with the same aspect ratio can bend up to 52.7° (0.92 rad).

Volumetric work density ($W_v$, the output work per unit volume) is a key parameter to evaluate the performance of actuator devices, which is directly associated with the strain and elasticity ($Y$, young's modulus) of materials by $W_v = \frac{1}{2} Y \varepsilon^2$. The colossal $Y$ value of $VO_2$ single crystals (140 GPa) is a great advantage compared to other actuation materials[31,32]. As shown in Fig. 4f, the SCVAs can produce a very large volumetric work density of up to 19.3 J cm$^{-3}$, which is comparable with those of shape memory alloys (SMAs)[31], and

superior to those of conventional $VO_2$ bimorph actuators[33–35], polymer actuators[36–39], and ferroelectric/piezoelectric (FE/PE) oxides-based actuators[40,41]. In addition, the work speed and stability of as-grown M2-R SCVAs and T-M2 SCVAs are examined by laser pulses at room temperature (for details refer to "Methods" section). The cut-off frequency (−3 dB attenuation frequency) of M2-R SCVA is higher than 5 kHz while this value for the T-M2 SCVA is ~1.8 kHz as shown in Supplementary Fig. 14a. This supports its competitive performance with high-speed output compared with other materials (Fig. 4f). The work speed of SCVAs is controlled by many factors, such as the gas pressure, ambient temperature, and speed of solid-solid phase transitions. As discussed in Fig. 2d, T → M2 transition is not a typical first-order transition like M2-R or M1-R transitions, which results in the relatively slow work speed of T-M2 SCVAs. Moreover, both M2-R SCVA and T-M2 SCVA show no degradation on $\theta_{max}$ over $1 \times 10^7$ oscillation cycles (Supplementary Fig. 14b), an ultra-stable output of SCVAs. This benefits from their simplest device structure and good chemical stability[27].

Energy conversion efficiency ($\eta$), the ratio of the output mechanical work to input heat, is another key factor for thermal actuators. As discussed, the output work is proportional to $\varepsilon^2$, while the input heat is used to increase the temperature of $VO_2$ (i.e., specific heat) and trigger its phase transition (i.e., latent heat). The calculated $\eta$ of as-grown M2-R SCVA, ~2.43% (for calculation see Supplementary Discussion), is several times larger than those reported for $VO_2$ (M1/R) actuators (~0.8%)[27,30] and T-M2 SCVAs (~0.69%). It is worth noting that the actuation performance of T-M2 SCVAs with a strain up to 0.75% is comparable to that of the reported for M1-R SCVAs with a strain of ~1%, while the latent heat of T-M2 transition (up to 300 cal mol$^{-1}$) is much smaller than that of M1-R transition (~1030 cal·mol$^{-1}$)[2,26]. Therefore, T-M2 SCVA is a promising candidate for the energy-saving thermal actuator that is a very appealing topic in modern micro-robotics. It is concluded that the as-grown M2-R SCVAs have approached the theoretical performance limits of $VO_2$-based actuators in terms of work density, energy conversion efficiency, work speed, and stability.

## Discussion

As demonstrated above, the ordered assembly of multi-phases and the asynchronized phase transitions in $VO_2$ beams (with the same or different phase transitions) fully utilize all the MIT-associated phases and their phase transitions at the single domain level, innovating the understanding and applications of S–T phase diagram. Accordingly, we would like to propose a concept of "phase transition route devices" (PTRDs) based on the single-crystalline $VO_2$ beams, where the spatially asynchronous phase transition routes and the competition between different coexisting $VO_2$ phases are expected to trigger the promising properties and impressive applications of $VO_2$. The SCVA family, as the prototypical PTRDs, show good performance and diverse functions, verifying their impressive advantages beyond traditional $VO_2$-based actuators. We anticipate the substantially enriched understanding of the S–T phase diagram and the advanced stoichiometry engineering strategy developed here would pave the way for more novel phase device applications with improved performance.

In summary, stoichiometry engineering was used to selectively stabilize all the three insulating phases (M1, T, M2) in single-crystalline $VO_2$ beams. The ability to spatially engineer phase inhomogeneity and phase transitions with stoichiometry gradient open opportunities for designing and controlling functional phases/domains of $VO_2$. As a typical device application, the

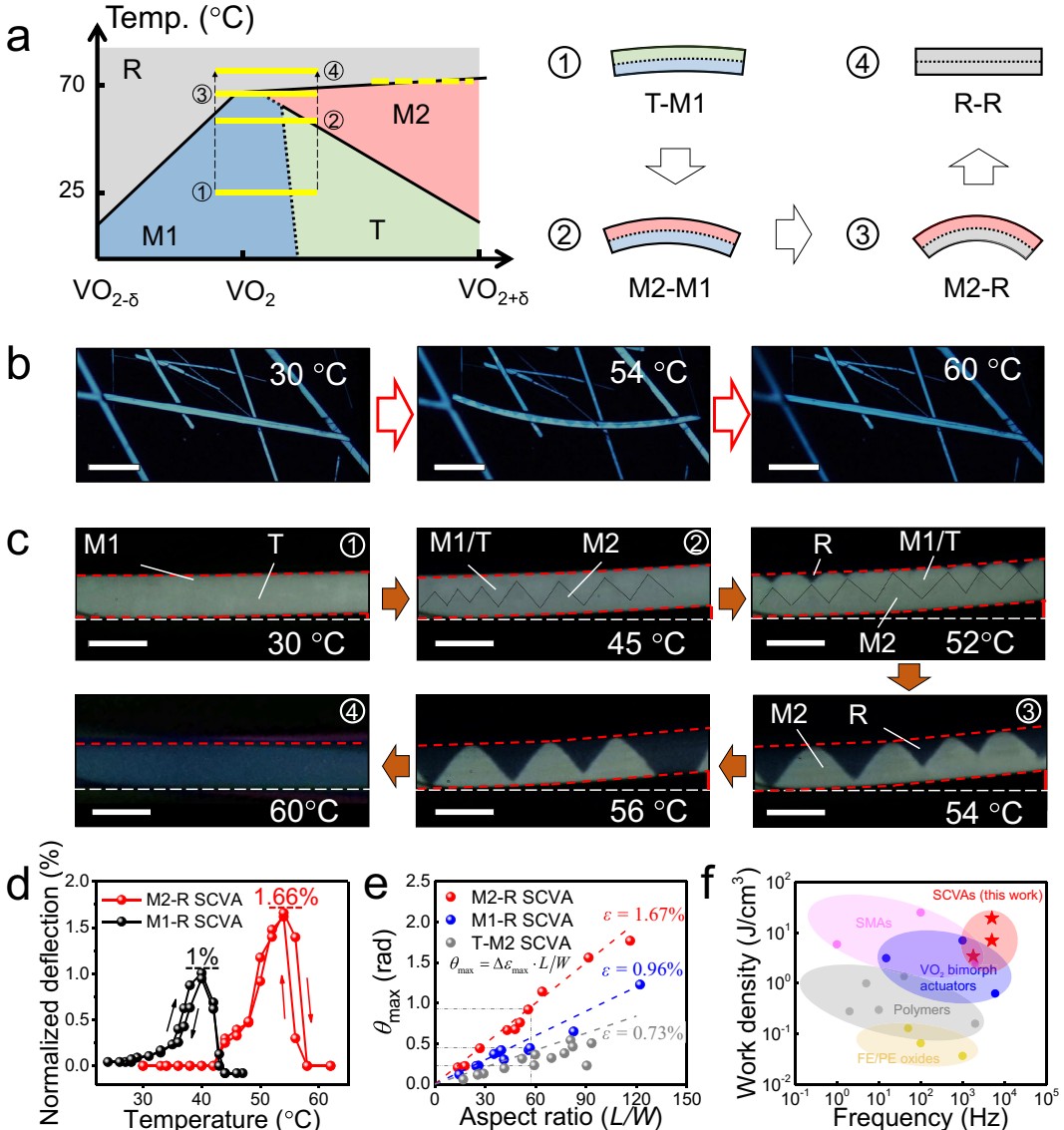

**Fig. 4 W-doped SCVA with superior actuation performance. a** Working mechanism for the W-doped M2-R type SCVA with the array of yellow gradient bars tracing the actuation process upon heating (top-right figures). **b** Optical images of a W-doped SCVA at increasing temperature (scale bars are 40 μm). **c** Temperature-dependent optical images of another SCVA showing a clear domain evolution process (scale bars are 5 μm), where black dash lines are plotted at the positions of domain walls between M2 and M1/T domains. The numbers at the top right-hand corner of the images correspond to the schematic images in Fig. 4a. **d** Temperature-dependent deflection of a M2-R type SCVA compared to a M1-R type SCVA. **e** Plots of $\theta_{max}$ of SCVAs versus their $L/W$ ratios. The colored dash-lines are theoretical $\theta_{max} - L/W$ plots of SCVAs utilizing the unidirectional strain ($\varepsilon$) of VO$_2$ across the phase transitions, $\varepsilon = 1.67\%$ for M2-R transition, $\varepsilon = 0.96\%$ for M1-R transition, and $\varepsilon = 0.73\%$ for T-M2 transition. **f** Actuation frequency and volumetric work density for various actuator systems, including ferroelectric/piezoelectric (FE/PE) oxides, polymers, shape memory alloys (SMAs), VO$_2$ bimorph actuators, and SCVAs.

SCVA family was demonstrated with attractive performance and functional diversity. As distinctly different physical properties are associated with these phases and phase transition routes, our work may provide possibilities to achieve collectively and internally tunable functionalities of MIT in VO$_2$.

## Methods

**Synthesis of VO$_2$ beams.** VO$_2$ beams were prepared via CVD reaction using 1–8 mg V$_2$O$_5$ powder (AR, 99.99%) as the V source uniformly mixed with SiO$_2$ (AR, particle size ~10 μm, 99%) with an appropriate mass ratio of 0.25–4. Through a typical reaction route, the oxide mixture was loaded onto a quartz boat at the center of the furnace and a rough amorphous quartz substrate (320 mesh) was placed over the precursor. The system was firstly heated up to 550 °C within 30 min and then slowly heated up to 850 °C at a heating rate of 10 °C min$^{-1}$. The system was kept at 850 °C for 1–120 min and then naturally cooled down to room

temperature, using Ar carrier gas (15 sccm) at 3 Torr pressure (in the whole process). The fabrication of W-doped VO$_2$ beams followed the same experimental procedure with 0.5–2.5 mg WO$_2$ powder placed aside the V$_2$O$_5$/SiO$_2$ mixture.

**Characterization.** The high-resolution transmission electron microscopy (HRTEM) images and the energy-dispersive X-ray spectroscopy (EDX) patterns were obtained using a FEI Talos F200X instrument. The Raman spectra of products and the pulse laser were obtained by using a HORIBA Raman spectrometer (LabRAM HR Evolution), with an excitation wavelength of 532 nm. Olympus optical microscope (BX 51) equipped with a charge-coupled device camera was used to capture the optical images. The electrical measurement was taken using a Keithley 4200-SCS semiconductor analyzer.

**Statistics of phase structure.** Within a selected area at the center of the substrate, 15 VO$_2$ beams with different lengths and widths were chosen for Raman

measurements. The Raman signals of three points of every beam were captured and analyzed, including the two ends and center of beams. The average value and standard error of measured $\omega_O$ phonon frequencies were calculated for every sample. Raman measurements were done at room temperature unless otherwise specified.

**Work speed and stability test**. First, the cantilevered SCVA was irradiated by a focused laser beam with an appropriate power to achieve its maximum deflection angle. Second, an optical chopper with a maximum frequency of 5 kHz was used to chop the incident laser at varying frequencies and to oscillate the actuator, and then captured by a CCD camera. Finally, the plots of maximum deflection angle versus pulse frequency and the cycle number of oscillations were respectively used to calculate/analyze the work speed and stability of the actuator.

## Data availability

The data that support the findings of this study are available from the corresponding author upon reasonable request.

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

## Acknowledgements

We acknowledge the support received from the National Natural Science Foundation of China (Grants Nos. 91963129 and 51776094), Basic Research Project of Science and Technology Plan of Shenzhen (Grant No. JCYJ20180504165655180), Foundation of Shenzhen Science and Technology Innovation Committee (Grant No. JCYJ20180302174026262). This work was also supported by SUSTech Core Research Facilities.

## Author contributions

R.S. and C.C. gestated the idea and designed the experiments and wrote the manuscript. R.S. grew the materials and performed main characterizations. Y.C. helped to analyze the actuation performance of SCVAs. X.C. and Q.L. helped to perform TEM and EDS characterization and analysis. Z.Z. and N.S. analyzed the results with R.S. And suggestions from N.W. and C.C. helped to explain the experimental results. A.A. justified the results and reviewed the interpretations and revised the manuscript. All the authors discussed the results and commented on the manuscript.

## Competing interests

The authors declare no competing interests.
