## [Peer Review File · Nature Communications]

REVIEWER COMMENTS

Reviewer #1 (Remarks to the Author):

The authors report the growth of phase controlled VO₂ nanowires with composition gradient. The use of SiO₂ as oxygen inhibitor is interesting. The development of self-actuation in VO₂ wires with composition gradient is interesting and novel. The experimental results are important. The manuscript could be dramatically improved if the authors can improve their presentations. I suggest the authors to address following comments.

- (1) Currently, the authors use optical contrast or EDX or Raman to check the composition gradient along the radial direction of the wire. They have their own merits but weakness. Could the authors image the wire via TEM along the radial direction for the sample at the temperature of interest? One can use selective area electron diffraction to reveal the local crystal structure.
- (2) It is known that the mechanical boundary conditions of the substrate influence the phase transition of VO₂. For example, different substrates give you different phase transition kinetics (J Cryst Growth 543, 2020, 125699). So if the wire is moved to a different location or substrate, could the authors still observe the identical self-actuation process?
- (3) I assume the “doper” is “dopant” in this sentence “Oxygen vacancies can work as electron dopers that stabilize the metallic R phase at low temperature”?
- (4) Please directly label the domains on the images of Fig. 1d. Currently, the contrast between M1/M2 is low.
- (5) The use of “inject/eject oxygen flow” may not be proper. The authors are suggested to use more proper terms to describe the process.
- (6) What is the mechanism of the inhibiting process of SiO₂? See “SiO₂ effectively inhibits the evaporation of V₂O₅ so that most of V sources are locally converted to VO₂ within the oxide mixture, while the other product, O₂, is released into reaction atmosphere...” How would SiO₂ inhibit the evaporation of V₂O₅?
- (7) In fig. 3d, why at the two sides of the nanowire there are more oxygen signals? For the case of vanadium, it seems to be intuitive.
- (8) Please label the domains in Fig. 4b clearly: for example, the authors can use dash line to outline certain phase regions.
- (9) In Fig. 4b, are the three images per column describe the same wire?

Reviewer #2 (Remarks to the Author):

The authors report an oxide-inhibitor-assisted CVD method of VO₂ beams growth for controlling their stoichiometry. A stoichiometry-temperature phase diagram is proposed to demonstrate different phase transition routes of VO₂ beams with different oxygen content. Furthermore, Single-crystalline VO₂ actuators (SCVAs) designed from different phase transition routes of non-stoichiometric VO₂ beams are demonstrated. Although the stoichiometry-temperature phase diagram and actuators for VO₂ has been known in previous literature, the method of controlling oxygen content of VO₂ during growth and the idea of utilizing different phase transition route to design VO₂-based actuators are very inspiring. I believe the work is worthy of publication at Nature Commun., after the following concerns /questions are addressed:

- 1) The stoichiometry-temperature phase diagram (Figure 1b) that proposed by the authors has already been known in the literature (e.g., Fig. 3A in “Defect-engineered epitaxial VO₂±δ in strain engineering of heterogeneous soft crystals” by Wang et al., Sci. Adv. 2018). Since the authors credit this diagram as their proposal, I was wondering if there is any new aspect from this proposed diagram as compared to previous work?
- 2) The method of controlling oxygen content in VO₂ beams by oxide inhibitor (SiO₂) is helpful for the VO₂ community. The EDS results (Figure S5b) confirm that VO₂ beam grown from this method is undoped. This finding, however, is a little bit surprising to the referee. For example, WO₂ and SiO₂ has similar melting point. VO₂ can be doped by WO₂ but not by SiO₂. Could the author provide more explanations in the underlying physics to help readers to understand?

3) The resistance-temperature curves (Figure S7) are provided to support the claim that as-grown VO₂ beams have high qualities (mentioned in line 21, page 13 of the main text). To make the claim more convincing, I suggest the authors provide resistivity-temperature data. Because besides the extend of MIT jump in resistance, it is also important for high-quality VO₂ in metal phase to have high conductivity (usually on order of 1E5 S/m for single-crystalline VO₂ nanowires/beams). Another issue is a large supercooling is shown in Figure S7. Could the authors comment on this?

4) In the latter half of the manuscript, the authors demonstrate SCVAs based on VO₂ with gradient oxygen content along their widths. However, the arrow line in Figure 1b seems to imply that sample 1-5 introduced at the beginning do not have oxygen gradience. It is not clear in the manuscript if the authors apply other controls in the experiment to achieve gradient VO₂ or all as-grown VO₂ beams are naturally gradient.

5) This manuscript needs to be further polished in writing to make it more precise and easier to read. For example, (1) adding scale bar in figure 1d, inset of 3c and 3d, etc.; (2) fixing a typo "insulating R" in line 20, page 1 (I suppose this to mean "insulating T"); (3) defining terms "reaction time" and "deposition time" in figure 2b-c's caption; (4) adding citation after "by referring to previous publications" in line 2-3, page 7; (5) adding figure index for clarity, such as changing "routes 1-5" to "routes 1-5 in figure 1b" (line 20, page 11) and etc.

Reviewer #3 (Remarks to the Author):

The authors present experimental results from a method to create stoichiometry engineered growth of VO₂ nanobeams. The oxygen gradient in the beams result in different phases and hence phase transitions within a single beam thus driving a self-bending actuation phenomenon.

The experimental results presented are clear and this is an extensive study. However, I do not think this work adds anything novel to the already available extensive literature on VO₂ actuators.

A few important things in this manuscript:

- (1) the CVD method to create stoichiometry engineered growth. The details are useful for the community working in oxide nanostructures.
- (2) phase stabilization and engineering of different phases in VO₂. This has been done through various methods and by different groups over the past 10 years. The manuscript contains many relevant references.
- (3) VO₂ actuators - again, the applications of phase transitions in VO₂ for thermochromic windows, actuators etc. has been studied extensively, including by these authors.

Based on the concerns in (2) and (3) above, I do not think the manuscript has enough novelty to warrant publication.

Response Letter

Response to Reviewer#1:

The authors report the growth of phase controlled VO₂ nanowires with composition gradient. The use of SiO₂ as oxygen inhibitor is interesting. The development of self-actuation in VO₂ wires with composition gradient is interesting and novel. The experimental results are important. The manuscript could be dramatically improved if the authors can improve their presentations. I suggest the authors to address following comments.

(1) Currently, the authors use optical contrast or EDX or Raman to check the composition gradient along the radial direction of the wire. They have their own merits but weakness. Could the authors image the wire via TEM along the radial direction for the sample at the temperature of interest? One can use selective area electron diffraction to reveal the local crystal structure.

Response: Thanks for this good suggestion. We agree with the reviewer that all the mentioned methods, including TEM which is always equipped with selective area electron diffraction (SAED), can be possibly used to check the composition gradient along the radial direction of the wire. Except SAED, we discussed the merits and weaknesses of these techniques in the text, including optical contrast, EDX and Raman, and characterized our samples by combining them. We concluded that the *in-situ* optical observation of the domain behavior assisted by Raman spectroscopy is the most convenient way to achieve the real-

time monitoring of phase transitions in VO₂ structures. These characterization methods mutually determined the composition gradient along the radial direction of the wire.

We also agree that the selective area electron diffraction is a powerful method to precisely characterize the crystal structure of nanomaterials. It should be an ideal method to link the oxygen nonstoichiometry (EDX data) with the phase structures of VO₂. However, SAED requires the thickness of the samples to be smaller than ~200 nm to allow electron beam passing through to form the measurable diffraction pattern; this requirement limits the application of SAED. We tried to check the structures of our samples using SAED, however, we found it's hard for such self-bending wires used in this work. This difficulty may originate from many aspects, including the large thickness of the samples, very close lattice structures of different VO₂ phases, possible perturbations from structural distortion or overlapping several domains with different phases or orientations at the bending part of the studied wire, and the inevitable heating effect of electron beam. In the following we summarize the unsuccessful results from SAED on our samples:

1) The relatively large thickness of the wires with oxygen gradient.

According to Supplementary Movie 1 and Supplementary Movie 2, it is noted that the lateral oxygen gradient exists in the VO₂ wires with a large radial size (usually > 1 μm), because their growth follows a unique wetting layer-assisted mechanism as discussed in Supplementary Discussion 4. If the diameter of VO₂ wires is comparable with the thickness of the wetting layer (hundreds of nanometers), the wires may be completely covered with the wetting layer even though the wetting layer is being consumed. As a result, the oxygen gradient is hardly produced in slim VO₂ wires.

We have presented the self-bending wires in Figure 3d using TEM (Figure R1), accompanied with EDX (Figure R1b, 1c) and SAED testing. According to the bright-field TEM images in Figure R1d, it is observed that the wire is rather too thick to allow the transmission of either light or electron; this results in completely black contrast in the TEM image. It is quite convenient to take the EDX line scanning along its radial direction and obtain the distribution of elements since the characteristic X-ray is emitted out in all directions, which can be easily collected by the detectors (Figure 1b, 1c). However, we failed to obtain appropriate SAED patterns from the VO₂ wires with an evident lateral oxygen gradient, i.e., the wire in Figure 3d, because it was too thick to allow the transmission and diffraction of electron beam.

Figure R1. (a) Optical images of a self-bending VO₂ beam (the one in Figure 3d) upon heating and its EDS mapping data in (b) and (c). (d) TEM images of the beam with the increasing dose of incident electron beam (Updated Supplementary Fig. 7c).

2) Quite similar lattice structures of the VO₂ phases

Despite the failure of using SAED for thick wires with oxygen gradient, we still thought the possibility to use SAED for relatively thin VO₂ wires to distinguish different phase structures.

We, thus, studied a thin VO₂ wire with the room-temperature-stable M2 structure via SAED; the SAED patterns at two usual zone axes of $[1\ 0\ -2]_{M2}$ and $[1\ 0\ -1]_{M2}$ are shown in Figure R2a as control. It is noted that the SAED patterns of M2 phase are completely overlapped with those of the typical M1 phase (Figure R2b). This result can be understood by considering the similar crystal structures of these phases with quite close periods ($d_R=5.700$, $d_{M1}=5.755$, $d_T=5.755\sim 5.797$ and $d_{M2}=5.797$, all in Å). The period differences are all < 0.1 Å, and the difference between M2 and T phase is < 0.05 Å, which is not possible for SAED to distinguish that.

The different forbidden diffraction points in the SAED patterns may be used to identify different VO₂ phases for the thick VO₂ wires (Nano Res., 2021, <https://doi.org/10.1007/s12274-021-3355-6>). We thus provide the simulated SAED patterns along three usual zone axes as shown in Figures R2c and R2d. It is discovered that one cannot distinguish VO₂ phases at zone axis 1 while R phase can be distinguished at the zone axis 2, and all these phases can be identified at the zone axis 3. However, the as-grown VO₂ wires, including those with the oxygen gradient, usually have rectangular cross-sections and always aligned with the longitudinal edge perpendicular to the zone axis 1, as reported in our previous publication (Sci. Rep., 2014, 4, 5456). In experiments, it is very hard for the TEM operators to tilt the wire by 45° towards the zone axis 3 from its preferential state that is normal to the zone axis 1, because the tilt range of TEM stage is usually limited to $\pm 30^\circ$.

In conclusion, it is difficult to apply SAED to identify the local crystal structure of VO₂ wires, because the tiny difference of the crystal lattices of VO₂ phases cannot produce measurable difference in the SAED patterns at the preferential zone axis of as-grown VO₂ wires.

Figure R2. SAED of single-crystalline VO₂ beams. (a) Experimental SAED patterns of a M₂ phase VO₂ beam at the zone axis $[1\ 0\ -2]_{M2}$ and zone axis $[1\ 0\ -1]_{M2}$. (b) Experimental SAED patterns of a M₁ phase VO₂ beam at the zone axis $[1\ -2\ 2]_{M1}$ and zone axis $[1\ -1\ 2]_{M1}$. (c) Schematic of as-grown VO₂ beams with three primary zone axes 1-3, normal to the axial direction (c_R) of VO₂ beam. (d) First-principle simulated SAED patterns of M₁, M₂ and R phase VO₂ single crystals at zone axes 1-3. (updated Supplementary Fig. 8)

3) Other possible issues

Generally, it is necessary for SAED to ensure that the sample is free of strain, otherwise the structural deformation could distort the final SAED patterns or break the original lattice symmetry. However, the suggested experiment that uses the SAED to characterize the laterally asymmetric lattice structure usually encounters a bending state which contains a complex strain distribution as well as a complicated domain pattern (at the sub-micrometer scale), as shown in Figure 3c. This means that the strained VO₂ part in the selected area aperture (>2 μm) may give a SAED pattern deviated from the theoretical one, with evident shift, elongation, and distortion of the diffraction spots, or the appearance of the forbidden reflections. Furthermore, the heating effect of electron beam cannot be ignored as shown in Figure R1d, where the irradiation of electron beam triggered the bending and phase transition of as-grown VO₂ wires because of their relatively low phase transition temperature. As a consequence, one can hardly distinguish local crystal structures from the SAED patterns because of all the mentioned barriers above.

In summary, we think that SAED may not be a proper method to characterize the unique phases of VO₂. We thank again for the reviewer to alarm us with this opportunity to analyze and compare different characterization tools for the structure study of VO₂ samples. The clarification on the weakness of SAED in our present case may help other scientists to advance their study by choosing appropriate characterization approaches, especially on the VO₂ phases related topics. Accordingly, we added this discussion and related data to Supplementary Discussion 3 for researchers in related fields.

(2) It is known that the mechanical boundary conditions of the substrate influence the phase transition of VO₂. For example, different substrates give you different phase transition kinetics (J Cryst Growth 543, 2020, 125699). So if the wire is moved to a different location or substrate, could the authors still observe the identical self-actuation process?

Response: This is a very good question. We agree that the mechanical boundary conditions of the substrate do influence the phase transition of VO₂. We thank the reviewer for the suggested article, to better understand the kinetics of phase transition. As reported by the article (J. Cryst. Growth 543, 2020, 125699), the strong interaction between the growth substrate c- Al₂O₃ and VO₂ could trigger the formation of a multi-domain configuration of VO₂ wires during the phase transition; similar phenomena have been well investigated (Nano Lett., 2006, 6, 2313; Nat. Nanotechnol., 2009, 4, 420). The unique oxide wetting layer-assisted growth behavior of VO₂ wires leads to their half-embedded growth on the oxygen-containing substrates, such as quartz, sapphire, and SiO₂/Si (J. Cryst. Growth, 2009, 311, 1571; Sci. Rep., 2014, 4, 5456).

By contrast, the VO₂ wires grown on h-BN undergo a rather fast phase transition, which is attributed to the weak nature of vdW bonding at the VO₂/h-BN interface and the reaction inert of h-BN. The above results suggest that the weak vdW force between VO₂ and substrates has a little effect on the phase transition kinetics of VO₂. So, if the wire is moved

to a different location or substrate, we should still observe the identical self-actuation process because the interaction between VO₂ wires and substrates are too weak to cause additional impact on their phase transitions.

We thus compared the self-actuation process of an as-grown free-standing VO₂ beam on a rough quartz substrate with that of the same one transferred to the edge of a flatten SiO₂/Si substrate (Figure R3). The optical images at increasing temperature indicated its consistent self-actuation processes accompanied with almost the same domain evolution behavior upon identical ambient temperature.

In conclusion, the self-bending actuation of as-grown VO₂ wires is independent of the supporting substrates or their locations that the VO₂ wires are transferred to, verifying that as-grown VO₂ wires are free-standing. We accordingly added this discussion to the Supplementary Discussion 7.

Figure R3. Temperature-dependent optical images of (a) a self-bending W-doped VO₂ wire on a rough quartz substrate, and (b) the one transferred to the edge of a flatten SiO₂/Si substrate. (updated Supplementary Fig. 15)

(3) I assume the “doper” is “dopant” in this sentence “Oxygen vacancies can work as electron dopers that stabilize the metallic R phase at low temperature”?

Response: Thank you very much for pointing out this misnomer. We replaced “doper” with “dopant” in page 7, line 2.

(4) Please directly label the domains on the images of Fig. 1d. Currently, the contrast between M1/M2 is low.

Response: As instructed, we labelled the domains in Fig. 1d as shown in Figure R4. Currently, the revised Figure R4 provides a better view for readers with the labelled domains. We accordingly updated Figure 1d.

Figure R4. Optical images of the domain evolution process of a VO₂ (T) beam upon heating (revised Figure 1d).

(5) The use of “inject/eject oxygen flow” may not be proper. The authors are suggested to use more proper terms to describe the process.

Response: We agree with the reviewer that the use of “inject/eject oxygen flow” may not be a proper term. We think that oxidation/deoxidation may be more suitable to describe the growth process in VO₂ beams under different oxygen partial pressure. Accordingly, we have revised the related contexts (page 10, line 13-14) and modified Figure 2a as shown in Figure R5.

Figure R5. Schematics of the CVD growth mechanism of VO₂ beams (revised Figure 2a).

(6) What is the mechanism of the inhibiting process of SiO₂? See “SiO₂ effectively inhibits the evaporation of V₂O₅ so that most of V sources are locally converted to VO₂ within the oxide mixture, while the other product, O₂, is released into reaction atmosphere...” How would SiO₂ inhibit the evaporation of V₂O₅?

Response: This is a very precious question. In our work, SiO₂ plays a crucial role in the growth of VO₂ and its phase modulation by acting as a novel inhibitor to control the release of V₂O₅ vapor and the oxygen partial pressure. Understanding the mechanism of inhibiting process of SiO₂ is generally essential to the researchers working on low-dimensional materials prepared by chemical vapor deposition. In the last submission, we provided a concise explanation in the main text and several related experimental results in SI. The reviewer’s comment makes us to realize the importance of this issue and clarify it further. Thus, we reorganized all the related results to clearly address the important role of SiO₂ as an inhibitor.

V₂O₅ is usually chosen as the vanadium source in the chemical vapor deposition (CVD) growth of VO₂ beams due to its relatively low melting point (Appl. Phys. Lett., 2012, 100, 103111), while its evaporation is quite fast and hard to control. Researchers have had a long-standing challenge to find an effective way to control the supply of gaseous precursors, specifically the vapor generated by heating solid sources. Recently, we developed the oxide inhibitor (OI)-assisted growth of two-dimensional MoS₂ monolayers (ACS Nano 2020, 14, 7593–7601), where the release of Mo vapor (MoO₃) can be finely modulated by inert oxide inhibitors. It is discovered that the thicker OI layer leads to the longer diffusion path of Mo vapor through the oxide mixture, and thus a larger portion of Mo source is trapped within the inhibitor layer. Inspired by this work, we developed a similar OI strategy to control the release of V₂O₅ vapor in the CVD growth of one-dimensional VO₂ beams, and achieved a good control of the V vapor supply as well as a continuous control of oxygen partial pressure in the reaction process.

Below, we discuss the working mechanism of SiO₂ inhibitor from physical and chemical points of view.

SiO₂ works as a physical trapping material to inhibit the fast release of V₂O₅ vapor:

We comprehensively study the reaction residues by using optical microscopy, XRD, and Raman etc., as shown in Figure R6-R8. The color of SiO₂ changes from white to yellow or orange (the color of V₂O₅) after the CVD reactions, shown in Figure R6b, suggesting that a considerable amount of V₂O₅ attaches to the surface of SiO₂; this is verified by Raman spectra (Figure R7). Moreover, XRD and Raman studies of reactant residues show that there are no additional compounds other than the reactants and typical products, including V₂O₅, V₆O₁₃, VO₂, and SiO₂ (Figure R7 and Figure R8b), suggesting that reaction inert of SiO₂ in the present reaction conditions. Therefore, we conclude that SiO₂ works as an effective physical trapping material to inhibit the fast release of V₂O₅ vapor.

SiO₂ promotes the conversion of V₂O₅ to VO₂, modulating the release of V₂O₅ vapor and oxygen partial pressure:

Even though SiO₂ does not directly participate with the reaction, it plays an essential role in the modulation of CVD reaction process. We studied the mass loss of reactants *vs.* deposition temperature and time at various additional amounts of SiO₂ as shown in Figure R8a (Supplementary Fig. 3c), as well as the characterizations of corresponding reaction residues. It is unexpected to see that at $T < 850$ °C, the V₂O₅ evaporation (indicated by mass loss) in the absence of SiO₂ is even slower than that with SiO₂ as shown in Figure R8a. This result can be attributed to the unique properties of V₂O₅. V₂O₅ powder tends to melt and merge into a single mass with a relatively low evaporation rate at $T < 850$ °C; at this period, the evaporation of V₂O₅ is self-limited. Once the temperature increases to 850 °C, the evaporation of V₂O₅ is accelerated. The optical images in Figure R6a verify the above deduction by showing that the residues in the absence of SiO₂ at $T < 850$ °C are glassy crystals while they change to powder-like substance at $T = 850$ °C. In addition, Raman spectra (Figure R7) indicate that at $T < 850$ °C the main component of the reaction residues is V₂O₅ while at $T = 850$ °C it contains V₂O₅ with a small amount of V₆O₁₃. This suggests that most of the vapor in the reaction atmosphere should be in the form of V₂O₅ but few O₂ is released from the precursor due to its limited reduction as shown in Figure R9a.

By contrast, the addition of SiO₂ breaks the self-limitation of V₂O₅ evaporation at $T < 850$ °C by dispersing V₂O₅ powder; dispersed V₂O₅ powder has a large exposed surface with enhanced evaporation by SiO₂ at $T < 850$ °C (Figure 8a). From the Raman spectra in Figure R7 and XRD analysis in Figure R8b, it is concluded that a large proportion of V₂O₅ source has been trapped and reduced to VO₂ during the reaction. This is accompanied with a large amount of O₂ released to the reaction atmosphere that results in a relatively high oxygen partial pressure as shown in Figure R9b. Therefore, the mass loss at $T = 850$ °C should be very limited considering the low evaporation rate of VO₂ and physical inhibition of SiO₂,

which is consistent with the result presented in Figure R8a. This claim can be also supported by the fact that there is no increase or growth of VO₂ beams at $T = 850\text{ }^{\circ}\text{C}$ in the SiO₂-assisted CVD reactions as shown in Supplementary Fig. 3a. As the reduction of remaining V₂O₅ keeps ongoing at $T = 850\text{ }^{\circ}\text{C}$, the partial oxygen pressure can be maintained at a relatively high level for the effective stoichiometry engineering of as-grown VO₂ beams.

The optical images in Figure R8c and Supplementary Fig. 3a further verify the reaction process by analyzing the final products. Obviously, at $T < 850\text{ }^{\circ}\text{C}$, the beams in the presence of SiO₂ grow faster than those with no SiO₂; this finding is in good agreement with the self-limited V₂O₅ evaporation in the absence of SiO₂. After the reaction at 850 °C for 20 min, with the increase of SiO₂ dosage, the nucleation density of VO₂ beams is greatly decreased. This is corresponded to the inhibited V₂O₅ evaporation in the presence of SiO₂ and the promoted evaporation without SiO₂ at $T = 850\text{ }^{\circ}\text{C}$.

Figure R6. Optical images of reaction residues of the oxide precursors during the CVD reaction in the (a) absence and (b) presence of SiO₂. The insets are the magnified view of the oxide residues (updated Supplementary Fig. 4).

Figure R7. Optical images (scale bars are 20 μm) and Raman spectra of oxide residues under: (a) 4 mg V_2O_5 at 750 $^\circ\text{C}$; (b) 4 mg V_2O_5 mixed with 4 mg SiO_2 at 750 $^\circ\text{C}$; (c) 4 mg V_2O_5 at 850 $^\circ\text{C}$; and (d) 4 mg V_2O_5 mixed with 4 mg SiO_2 at 850 $^\circ\text{C}$. The colored circles show the positions of measurements, linked to the Raman data with the same colors (updated Supplementary Fig. 5)

Figure R8. (a) Plots of mass loss of the oxide mixture during the CVD reaction with different amounts of SiO₂ added to the system, where the colored background shows the change of reaction temperature (updated Supplementary Fig. 3c). (b) XRD patterns of pristine SiO₂, the oxide mixture of SiO₂ and V₂O₅ before reaction, and the oxide residues after the reactions at different temperature. (c) Optical images of the products prepared by adding different amounts of SiO₂ to the system.

Figure R9. Schematics for the status of oxide precursors during the CVD reaction in the (a) absence and (b) presence of SiO_2 . (updated Supplementary Fig.6)

In summary, the inhibiting process of SiO_2 in the CVD growth of VO_2 beams is a joint result from physical inhibition and chemical inhibition. SiO_2 initially promotes the evaporation at $T < 850\text{ }^\circ\text{C}$ and then prohibits the evaporation at $T = 850\text{ }^\circ\text{C}$, accompanied with the increase in the oxygen partial pressure. We have accordingly revised the Supplementary information Discussion 2 to provide a better clarification for the role of SiO_2 inhibitor.

(7) In fig. 3d, why at the two sides of the nanowire there are more oxygen signals? For the case of vanadium, it seems to be intuitive.

Response: Thanks for this good question. We think the high oxygen signals at two sides of the wire should be attributed to the amorphous V_2O_5 layer as shown in the HRTEM image of Figure R10 (revised Supplementary Fig. 7a). It has been widely reported that the surface of VO_2 crystals is easily oxidized to V_2O_5 in air (Applied Surface Science 2014, 321, 464–4); the thin V_2O_5 layer can prevent further oxidation (The Journal of Physical Chemistry C, 2017, 121, 24877-24885). In addition, we found similar EDS element mapping results in Chem. Mater. 2019, 31, 699–706, where the O signals at the two sides of VO_2 nanorods are

stronger than that of the middle part. And, because the V_2O_5 layer is thin and amorphous, the bending behavior of oxygen gradient VO_2 wires is hardly affected.

Despite the unexpected strong O signals at the sides, the existence of oxygen gradient can be well confirmed from the mapping by comparing the distribution of oxygen intensity to vanadium intensity. As shown in Figure 3d, the left part of the beam has a higher V intensity and lower O intensity than the right part, verifying the stoichiometry difference along the lateral direction VO_2 beam.

Figure R10. HRTEM images of non-stoichiometric VO_2 beam, scale bar = 5 nm (updated Supplementary Fig. 7a).

To eliminate the surface oxidation or attached V_2O_5 layers, we washed as-grown VO_2 beams by 2M Na_2CO_3 aqueous solution for 10 h (Communications Materials, 2020, 1, 28), and analyzed the element distribution of the washed self-bending VO_2 beam. It is rather noticed that the strong O signals at the two sides disappeared as shown in Figure R11. We have added the related discussion to the main text in page 17, lines 17-22 and Supplementary Discussion 3.

Figure R11. EDS mapping of (a) O and (b) V along a washed T-M2 type SCVA. The white arrow shows the scan line along the lateral direction. (c) Element distribution of V and O across the lateral direction. (updated Supplementary Fig. 7d and 7e)

(8) Please label the domains in Fig. 4b clearly: for example, the authors can use dash line to outline certain phase regions.

Response: As suggested, we labeled the domains in Figure 4b and added some dash lines to locate the domain walls between M2 regions and M1/T regions as shown in Figure R12. As explained in the main text, the optical contrast between M1 and T phase is too weak to be distinguished by optical microscopy, and thus, it is hard to label them properly. In addition, we uploaded the Supplementary Movie 3 to show the domain evolution process of Figure 4b, which may provide the readers more information than the separate optical images in Figure 4b. Additionally, we also used dash line to outline the phase regions in the inset of Figure 3c.

(9) In Fig. 4b, are the three images per column describe the same wire?

Response: We thanks the reviewer to point out the improper form of image presentation. The top three images describe the self-bending process of a wire upon heating, while the bottom six enlarged images belong to another self-bending wire that demonstrate a relatively clear domain evolution process during heating. The two wires are from the same batch of samples and demonstrate similar actuation behavior. To avoid misunderstanding, we revised Fig. 4 as shown in Fig. R12. Several arrows were added to the updated Fig. 4b and Fig. 4c to guide the reader for a better view of the phase/domain evolution upon heating. In addition, we carefully checked the image and revised the presentation by: (1) changing the color of the gradient bar in Fig. 4a, and (2) linking Fig. 4c to the schematic images in Fig. 4a.

Figure R12. The revised Figure 4.

Response to Reviewer#2's comments:

The authors report an oxide-inhibitor-assisted CVD method of VO₂ beams growth for controlling their stoichiometry. A stoichiometry-temperature phase diagram is proposed to demonstrate different phase transition routes of VO₂ beams with different oxygen content. Furthermore, Single-crystalline VO₂ actuators (SCVAs) designed from different phase transition routes of non-stoichiometric VO₂ beams are demonstrated. Although the stoichiometry-temperature phase diagram and actuators for VO₂ has been known in previous literature, the method of controlling oxygen content of VO₂ during growth and the idea of

utilizing different phase transition route to design VO₂-based actuators are very inspiring. I believe the work is worthy of publication at Nature Commun., after the following concerns/questions are addressed:

1) The stoichiometry-temperature phase diagram (Figure 1b) that proposed by the authors has already been known in the literature (e.g., Fig. 3A in “Defect-engineered epitaxial VO₂±δ in strain engineering of heterogeneous soft crystals” by Wang et al., *Sci. Adv.* 2018). Since the authors credit this diagram as their proposal, I was wondering if there is any new aspect from this proposed diagram as compared to previous work?

Response: We thank the reviewer to point out our inappropriate statement. The stoichiometry-temperature (ST) phase diagram (Figure 1b) is a schematic diagram, which is constructed by referring to the existing publications, such as *Sci. Adv.* 2018, **4**, eaar3679, the example given by the reviewer, *Nano Lett.* 2011, **11**, 1443–1447, and *Nano Lett.* 2012, **12**, 6198–6205. This ST phase diagram is used to help readers to understand the phase transition processes of nonstoichiometric VO₂ and the working mechanism of the VO₂ devices utilizing the asymmetric phase transition routes, namely phase transition route devices that should be first proposed in this work. We did not intend to credit this phase diagram as our new proposal with the statement “..., by referring to previous publications and excluding the effect of stress.” To address the reviewer’s concern and prevent the misunderstanding, we updated this statement, in page 7, line 5-8, to a clearer format of: “According to previous publications^{10,21,22}, a classical schematic stoichiometry-temperature (S-T) phase diagram is supplied in Figure 1b as a reference to describe the impact of stoichiometry on VO₂ phase structures and the mutual revolution of these phases.” In addition, we revised the misleading phrase “proposed phase diagram” as “ST phase diagram” or “classical stoichiometry-temperature phase diagram”. We sincerely appreciate the reviewer to point this out, and we hope these revisions can properly cover this comment.

By the way, we think our work helps to experimentally verify this schematic ST diagram by providing a wide range of nonstoichiometric examples, enabling the comprehensive verification of its reliability and qualitative amendments. We found that the slopes of phase boundaries in previous publications are somewhat inconsistent. In the present work, according to our meticulous observation on the phase transition routes of VO₂ with different oxygen contents, and the self-actuation performance of SCVAs, M1-R, M1-T, T-M2, and M2-R, we can experimentally evaluate the slopes of different phase boundaries as Slope_{M2-R} < Slope_{T-M2} < Slope_{M1-R} < Slope_{M1-T}. That is to say, the M2-R phase boundary is rather flat, and the M1-T phase boundary is quite sharp.

In addition, in this work, we propose a new concept of “phase transition route devices” (PTRDs) based on the single-crystalline VO₂ beams, where the spatially asynchronous phase transition routes and the competition between different coexisting VO₂ phases are expected to trigger new properties and impressive applications. Single-crystalline VO₂ actuator

(SCVA), driven by the laterally asymmetric phase transition routes, is one of the typical “PTRDs”. Compared to the conventional VO₂ bimorph actuators, SCVAs have much simpler device structure, competitive actuation performance, and most importantly, superior stability. Therefore, it is believed that the idea of SCVAs paves a new path for further developments of advanced mechanical devices in near future. We think our work strengthens the in-depth understanding and utilization of the S-T phase diagram. To cover all the above points, we revised the Discussion section in page 22, line 20-22 and page 23, line 1-20.

2) The method of controlling oxygen content in VO₂ beams by oxide inhibitor (SiO₂) is helpful for the VO₂ community. The EDS results (Figure S5b) confirm that VO₂ beam grown from this method is undoped. This finding, however, is a little bit surprising to the referee. For example, WO₂ and SiO₂ has similar melting point. VO₂ can be doped by WO₂ but not by SiO₂. Could the author provide more explanations in the underlying physics to help readers to understand?

Response: Thanks for this valuable comment. It has been considered that SiO₂ is an inert oxide and cannot participate in the CVD growth of crystals, at least for the reactions conducted at $T < 1000$ °C. Therefore, researchers always use quartz (SiO₂) substrates, quartz boats and quartz tubes to fabricate the VO₂ single crystals and other materials, including two-dimensional transition metal dichalcogenides etc. The Si-doping in these materials has been rarely reported, but we agree with the reviewer on the possible doping issue with SiO₂, which is always taken for granted and ignored. We thank the reviewer to give us this opportunity to consider this point and, therefore, would like to provide the following explanations.

As mentioned by the reviewer, the melting point of WO₂ (1500-1600 °C) is close to that of SiO₂ (~1700 °C). However, rather the melting point, the vapor pressure of oxides is the factor that really matters in the CVD reactions. According to Nature 1949, **163**, 601–602, the tungsten oxides (WO₃, WO₂) are volatile above 800-900 °C, with a vapor pressure of ~0.2 Torr at 1023 °C, while the vapor pressure of SiO₂ at ~1027 °C is measured as 10⁻¹³ Torr (Samsonov G V. The oxide handbook. Springer Science & Business Media, 2013). It is concluded that WO₂ is a volatile oxide at high temperature especially under low pressure while the volatility of SiO₂ is nearly negligible.

Secondly, we need to discuss another question whether VO₂ can be doped by Si, considering that the VO₂ beams are grown on the quartz substrates. We searched related publications about Si-doped oxides and found that the Si-doping in metal oxides should be achieved by using highly active Si precursors, such as Si[N(CH₃)₂]₄ in atomic layer deposition (Appl. Phys. Lett. 2013, **103**, 192904) and tetraethyl orthosilicate (TEOS) in CVD (RSC Adv., 2017, 7, 10806-10814). By contrast, solid SiO₂ is chemically very stable so huge energy is required to break the strong covalent Si-O bonds and seize the Si from the

SiO₂ substrate. In addition, the substitutional doping of Si in VO₂ lattices seems to be thermodynamically unstable considering the large differences between SiO₂ and VO₂ in symmetry, bond type, bond length, etc. Consequently, we believe that the Si doping of VO₂ on the SiO₂ substrate may not happen at a temperature of 850 °C.

In conclusion, the nonvolatility of SiO₂ powder and the good stability of quartz substrate at 850 °C make the Si doping hard to achieve in the VO₂ lattice. This discussion can also be used to explain the absence of Si doping in other materials grown in the presence of SiO₂. We added this discussion to the Supplementary Discussion 6.

3) The resistance-temperature curves (Figure S7) are provided to support the claim that as-grown VO₂ beams have high qualities (mentioned in line 21, page 13 of the main text). To make the claim more convincing, I suggest the authors provide resistivity-temperature data. Because besides the extend of MIT jump in resistance, it is also important for high-quality VO₂ in metal phase to have high conductivity (usually on order of 1E5 S/m for single-crystalline VO₂ nanowires/beams). Another issue is a large supercooling is shown in Figure S7. Could the authors comment on this?

Response: Thanks for your kind suggestion. We calculated the resistivity data for the M1, T, and M2 VO₂ beams in Supplementary Fig. 10, the updated plots are shown in Figure R13. We found that the measured conductivities of metallic VO₂ beams are up to ~2 E4 S/m, which were slightly lower than the suggested value for the VO₂ single crystals with the order of 1 E5 S/m. The depressed electrical conductivity can be attributed to many factors, such as increased oxygen content in the lattice (J. Appl. Phys., 1974, 45, 2201-2206), inevitable contact resistance in two-terminal devices, and imperfect contact caused by the strain/stress of VO₂ beams across their phase transitions. Therefore, it is hard to evaluate the quality of as-grown VO₂ crystals by simply analyzing their electrical properties. For an accurate description of experimental results, we revised the corresponding statements in page 14, lines 1-4: “Furthermore, the electrical measurements are conducted to evaluate the phase transition properties of the VO₂ beams with room-temperature-stable M1, T, and M2 phases as shown in Supplementary Fig. 10. All the beams undergo colossal resistivity changes with 4-5 orders of magnitude across MIT, implying the good electrical switching performance of as-grown VO₂ beams.”

As mentioned by the reviewer, the hysteresis has been widely observed in the metal-insulator transition of VO₂, upon supercooling during the R→M (M1 and M2) transition. We discussed the role of external stress of VO₂ beams during the electrical measurements in Supplementary Discussion 5. It is noted that the VO₂ beams are subjected to a tensile stress during heating while they are under a compressive stress upon cooling. The asymmetric stress along the VO₂ beams should cause the asymmetric phase transition kinetics: the tensile strain slightly increases the phase transition temperature while the compressive strain

reduces the phase transition temperature, according to the well-investigated strain-temperature phase diagram (Nature Nanotechnology, 2009, 4, 732-737; Nature, 2013, 500, 431-434). In addition to the electrical measurements, we also observed the evident supercooling of free-standing VO₂ beams via optical microscopy and Raman spectroscopy, which indicated that the external stress might not be the only governing factor. Fan et al. (PHYSICAL REVIEW B, 2011, 83, 235102) reported different phase transition mechanisms between the M→R transition during heating and the R→M transition during cooling by analyzing their nucleation processes. They attributed the supercooling of R→M transition in free-standing VO₂ beams to the limited concentration of point defects that can work as the nucleation site of R→M transition but not in the M→R transition. Fan et al. reported a hysteresis width of ~13 °C in a freestanding VO₂ (M) nanowire, $T_{M\rightarrow R} = \sim 69$ °C and $T_{R\rightarrow M} = \sim 56$ °C, very close to our results. Therefore, the large supercooling may indicate that the CVD-grown VO₂ beams have relatively few defects.

The related discussion was added to the Supplementary Discussion 5. Thanks again for this precious comment from which we improved the presentation of some important results and provided necessary discussion accordingly.

Figure R13. Temperature-dependent resistivity plots of room-temperature-stable (a) M1 phase, (b) T phase, and (c) M2 phase single VO₂ beams. (updated Supplementary Fig. 10)

4) In the latter half of the manuscript, the authors demonstrate SCVAs based on VO₂ with gradient oxygen content along their widths. However, the arrow line in Figure 1b seems to imply that sample 1-5 introduced at the beginning do not have oxygen gradient. It is not clear in the manuscript if the authors apply other controls in the experiment to achieve gradient VO₂ or all as-grown VO₂ beams are naturally gradient.

Response: We thank the referee to raise this question. The arrow lines in Figure 1b do show that introduced samples 1-5 should not have the oxygen gradient, and they are only used to demonstrate the phase transition routes of VO₂ with different oxygen contents. The difference between the samples with the oxygen gradient (SCVA devices) and those without might be confused. It seems that it is necessary to clarify their differences and demonstrate

their relativity. To respond the referee's question, we would like to supply the following discussion.

We did not apply any specific control to achieve the VO₂ samples with oxygen gradient in the SiO₂ inhibitor-assisted growth strategy, while we found that the formation of oxygen gradient was affected by the diameter of VO₂ beams, their positions in relation to the substrate, and W-doping. According to our experimental observations, every batch of product contains both the gradient VO₂ beams and the uniform VO₂ beams (without oxygen gradient). The proportion of the beams with oxygen gradient varies in the range of 20-95%, as shown in the Supplementary Movie 1 (20-60%, T-M2 SCVAs in the undoped sample) and Supplementary Movie 2 (40-95%, M2-R SCVAs in the W-doped sample). It is noted that the wide VO₂ beams prefer to bend driven by the oxygen gradient, while the narrow beams (< 500 nm in diameter) often keep straight upon the change in temperature. In addition, the free-standing VO₂ beams are more likely to have oxygen gradient than the VO₂ beams lying flat on the substrate. It is interesting that the W-doping helps to increase the ratio of the gradient VO₂ beams and enlarge the oxygen gradient in VO₂, which is applied to fabricate M2-R SCVAs.

The wetting-layer mechanism in Supplementary Discussion 4 gives a good explanation for the above results, where the spatially uneven distribution of the wetting layer is believed to help the formation of the oxygen gradient. If the diameter of thin beams is comparable with the thickness of the wetting layer (at hundreds of nanometers), they may be always completely covered during the reaction, thus, the limited oxygen gradient will be created. When the VO₂ beams lay flat on the substrate, they are more likely to be completely immersed in the aggregated wetting layer on the substrate than the suspending ones. For the role of W-doping in the formation and increase of oxygen gradient in VO₂, it is believed that the reaction between WO₂ and VO₂ triggers the formation of oxygen vacancies in as-grown VO₂ beams, while this process is further influenced by the wetting layer. Based on the above understanding, it is possible to control the interaction between VO₂ beams and the wetting-layer during the CVD reactions for the robust spatial phase engineering of VO₂.

To cover this comment, we added a statement in page 17, lines 7-10: "It is discovered that 20-60% of the VO₂ beams prepared by the inhibitor-assisted CVD method can work as T-M2 SCVAs as shown in Figure 3b and Supplementary Movie 1. This observation demonstrates their straight-bending-straight bidirectional actuation capability upon heating as predicted in Figure 3a." To explain the situation and satisfy the concern of the reviewer, we added the following content to page 20, lines 2-3: "It is discovered that 40-95% of the as-prepared VO₂ beams can work as M2-R SCVAs, indicating that W doping favors the formation of stoichiometry gradient in VO₂ beams". In addition, we added the related discussion in the updated Supplementary Discussion 4 for a clear presentation.

5) This manuscript needs to be further polished in writing to make it more precise and easier to read. For example, (1) adding scale bar in figure 1d, inset of 3c and 3d, etc.; (2) fixing a typo “insulating R” in line 20, page 1 (I suppose this to mean “insulating T”); (3) defining terms “reaction time” and “deposition time” in figure 2b-c’s caption; (4) adding citation after “by referring to previous publications” in line 2-3, page 7; (5) adding figure index for clarity, such as changing “routes 1-5” to “routes 1-5 in figure 1b” (line 20, page 11) and etc.

Response: Thanks very much for these constructive comments. As instructed, we thoroughly revised the manuscript.

Response to Reviewer #3:

The authors present experimental results from a method to create stoichiometry engineered growth of VO₂ nanobeams. The oxygen gradient in the beams result in different phases and hence phase transitions within a single beam thus driving a self-bending actuation phenomenon.

The experimental results presented are clear and this is an extensive study. However, I do not think this work adds anything novel to the already available extensive literature on VO₂ actuators.

A few important things in this manuscript:

(1) the CVD method to create stoichiometry engineered growth. The details are useful for the community working in oxide nanostructures.

(2) phase stabilization and engineering of different phases in VO₂. This has been done through various methods and by different groups over the past 10 years. The manuscript contains many relevant references.

(3) VO₂ actuators - again, the applications of phase transitions in VO₂ for thermochromic windows, actuators etc. has been studied extensively, including by these authors.

Based on the concerns in (2) and (3) above, I do not think the manuscript has enough novelty to warrant publication.

Response: We would like to express our thanks to the reviewer for taking time to review our work. We are glad that the reviewer found some important things in this manuscript and

thinks some of them are useful to the community working on oxide nanostructures. We understand the concern on the novelty of our work, so we are encouraged to clarify and highlight the new aspects of this work and its substantial advances to the VO₂ field and comparing them with the most recent related topics in literature.

The reviewer mentioned that we have provided a detailed literature review in the manuscript, and it is our responsibility to introduce a comprehensive background of this work to the readers, especially some important representing articles in the related field. Moreover, there are still several critical gaps to fill according to the existing publications, i.e., the phase engineering and manipulation of different VO₂ phases and their advanced applications. In this work, we committed to face these challenges and provide an effective strategy to address them, along with presenting our exciting breakthroughs.

We think the negative concern on the novelty is originated from our poor presentation. Therefore, we enhanced the presentations to ensure the improved accuracy of the revised manuscript and focused on unique findings in the VO₂ field. We hope the revised manuscript and our point-by-point responses can satisfy the reviewer. We do welcome any further comments. We would greatly appreciate if the reviewer could take time to review our responses to the concerns in (1)-(3) which clarify the novelty of this manuscript, new aspects compared to the existing publications, and substantial advances in VO₂ field.

(1) the CVD method to create stoichiometry engineered growth. The details are useful for the community working in oxide nanostructures.

Response: Thanks for the comment. Although the CVD method has been widely used to grow oxide nanostructures, the controllable stoichiometry engineered growth has been rarely reported. In our work, we achieved a controllable stoichiometry modulation of VO₂ beams by introducing SiO₂ as an oxide inhibitor in the CVD reactions, where releasing V₂O₅ vapor and oxygen partial pressure are finely controlled (for more details, please refer to Supplementary Discussion 2). This work provides a simple and effective strategy to create stable and controllable reactants supply, which we believe can be referenced by the community working in oxide nanostructures, as mentioned by the reviewer.

(2) phase stabilization and engineering of different phases in VO₂. This has been done through various methods and by different groups over the past 10 years. The manuscript contains many relevant references.

Response: We agree that the phase stabilization in VO₂ has been achieved by several methods, as introduced in this manuscript, over the past 10 years. However, the basic understanding, manipulation methods, and applications of these phases are yet limited. Below, please find our detailed discussion:

Several strategies including chemical doping, external strain, and oxygen non-stoichiometry (the focus of this manuscript) have been applied for the stabilization and engineering of different VO₂ phases.

It has been widely reported in the last century that, the trivalent metal (Al³⁺, Cr³⁺, etc.) doping can trigger the formation of metastable T/M2 phase, while the hexavalent/pentavalent metal ions (W⁶⁺, Nb⁵⁺, etc.) help to stabilize the R phase. Some light doping atoms (H, B, Be, etc.) can work as interstitial dopants and effectively stabilize the R phase. The exact role of these dopants in the phase transition of VO₂ is yet unclear, since the metal doping simultaneously causes the structural distortion and the change of free electron density. Chemical doping of single element can either stabilize the R phase or T/M2 phase, but the selective stabilization of all VO₂ phases (M1, T, M2, R) in single doping systems is not achievable.

Strain engineering along the *c*_R direction is an efficient way to fully engineer the phase structure of single-crystalline VO₂ beams, the tensile strain stabilizes the M2/T phase while the compressive strain stabilizes the R phase. However, the strained VO₂ beams have higher energy than the free ones, which means the unstable state and limited lifetime of strained VO₂ in practical devices. There is another major challenge to locally/precisely implement the strain on VO₂ beams for their desired phase structure and phase transition properties in practical applications.

Moreover, the phase engineering of VO₂ and the in-depth study of its metastable phases are significant to explore more information about its MIT mechanism. The competition between the electron correlation-dominant Mott-Hubbard model and structural distortion-dominant Peierls model has confused the researchers on phase transition mechanism of VO₂ for decades. For an intensive investigation of the MIT mechanism, it is necessary to introduce a controllable perturbation to the VO₂ lattice excluding the large structural distortion. Therefore, the chemical doping and strain engineering seem not to be ideal choices.

In recent years, stoichiometry engineering has been regarded as a moderate way to stabilize metastable phases of VO₂. There have been several publications revealing the fact that oxygen vacancies reduce the MIT temperature, and the excessive oxygen stabilizes T/M2 phases. However, the full phase stability and phase engineering in VO₂ is far from well-investigation, in particular the spatial phase engineering of single-crystalline VO₂ beams.

For example, Zhang et al. (*Nano Lett.* 2011, **11**, 1443–1447) creatively fabricated the oxidized T/M2 VO₂ phases by injecting the oxygen flow into the CVD tube in the first 15 min of temperature ramping stage, however, only 15% of the final products were oxidized properly, implying the difficulty of oxidation control during the reaction. It is assumed that the low fraction of targeted samples is from the complex oxidation process of VO₂ that is

very sensitive to the oxygen partial pressure and reaction temperature. Therefore, a new protocol for the control of oxygen is highly required during the CVD reaction.

Wang et al. (Sci. Adv., 2018, **4**, eaar3679) and Kim et al. (Nano Lett., 2016, **16**, 4074–4081) developed similar template-induced CVD growth of M2/T phase VO₂ nanobeams on specific r-cut sapphire substrates. The V₂O₅ template played an essential role in the constrained oxidation of VO₂ beams in their cases. However, the oxygen content of the VO₂ beams was not finely engineered. In Wang et al.'s work, the stoichiometry of VO₂ nanobeams was only related to their alignments, wherein the vertical nanowires were in M2 phase and the horizontal nanowires were T phase, implying that this work did not carry the precise control of oxygen.

Kim et al. well linked the phase structure of VO₂ nanowires with their diameters: with the increase of diameter, the nanowires transformed from M1 to T to M2 phase. But they attributed various phase structures to the size-dependent internal strain of VO₂ nanowires from their lattice mismatch with the substrate, instead of the oxygen non-stoichiometry. In addition, the sizes of VO₂ beams prepared by these methods (up to 30 μm) were usually much smaller than those of the conventional CVD-grown VO₂ beams due to their complicated reaction conditions and confined nucleation.

Therefore, we found that the controllable stoichiometry engineering of single-crystalline VO₂ beams has not been achieved and efforts to modify the size of nonstoichiometric VO₂ crystals are greatly desired.

As you may have noticed, we have addressed the existing challenges through the presented oxide inhibitor-assisted CVD growth strategy in this work. We can selectively stabilize all the three insulating phases (M1, T, M2) in single-crystalline VO₂ beams by simply adjusting the composition of precursors and thus precisely modulating the oxygen partial pressure of the reaction system. In addition, the presented method enables the growth of VO₂ beams with large sizes at the sub-millimeter scale in length. More attractively, the laterally asymmetric phase transition routes in as-prepared VO₂ beams demonstrate that this method enables the spatial manipulation of their phase structure at the domain level (submicron scale), which have not been reported by any other groups till now.

Therefore, this work provides a powerful way to achieve multiphase stabilization and engineering in VO₂ systems, paving a new path to explore the underlying mechanism of the controversial MIT. Moreover, the single-crystalline actuators constructed in this work show outstanding actuation performances, including competitive work density, high work speed, long lifetime, and high energy conversion efficiency, verifying the powerful application potential of the engineered VO₂ systems. It is believed that this work can also inspire the study of other strongly correlated systems, especially in new phase transition physics and phase transition-driven applications.

Through the above comparisons and discussion, we believe that the phase management in single-crystalline VO₂ beams presented in this work is not simply an extensive work mimicking the previous publications. The ability to selectively stabilize all the three insulating phases (M1, T, M2) in high aspect ratio single-crystalline VO₂ beams followed by the novel phase spatial engineering is the first time to be demonstrated.

(3) VO₂ actuators - again, the applications of phase transitions in VO₂ for thermochromic windows, actuators etc. has been studied extensively, including by these authors.

Response: We agree that VO₂ actuators have been studied extensively while most of them are based on bimorph structures, since initial reports in 2010 (J. Appl. Phys., 2010, 107, 074506 and J. Appl. Phys., 2010, **108**, 083538).

Taking good of the strengthened understanding of the S-T phase diagram and the spatial phase manipulation of VO₂ beams in this work, we propose a new concept of “phase transition route devices” (PTRDs) based on the single-crystalline VO₂ beams, where the spatially asynchronous phase transition routes and the competition between different coexisting VO₂ phases trigger new properties and impressive applications. Single-crystalline VO₂ actuator (SCVA), driven by the laterally asymmetric phase transition routes, is one of the typical “PTRDs”. Compared to the conventional VO₂ bimorph actuators, SCVAs have much simpler device structure, competitive actuation performance, and most importantly, superior stability. Therefore, it is believed that the idea of SCVAs paves a new path for further developments of advanced mechanical devices in near future.

The concept of SCVA and the prototype device were firstly reported in our recent work (Adv. Funct. Mater., 2019, **29**, 1900527), where only M1-R type SCVA was demonstrated because of the lacking methodology to stabilize/manipulate other oxidized VO₂ phases (T and M2). In the present work, based on the S-T phase diagram, we propose all four possible types of SCVAs of the SCVA family. The effective phase management in single crystal beams enables three types of SCVAs with evident deflection outputs. Most importantly, we have obtained the highest theoretical performance of VO₂ based actuators by utilizing the colossal uniaxial strain of M2-R transition ~1.67%. The M2-R SCVA demonstrates a rather high work density of ~19.3 J·cm⁻³, which is comparable with SMAs, and simultaneously promises an outstanding response speed (> 5 kHz). In addition, the ultra-stable device structure and reliable actuation performance guarantee the high-efficiency work of SCVA family in various working environments.

Therefore, we believe that the phase management method of single crystalline VO₂ beams in this work greatly expands the concept of SCVAs by building the whole family of SCVAs. Furthermore, the concept of SCVA demonstrates substantial advances to VO₂ actuators as well as the advanced VO₂ applications driven by engineered phase transition routes.

Below, there are some detailed or auxiliary explanations about the unique fingerprint of this work in VO₂ actuators.

a) Advantages of SVCAs compared to traditional bimorph actuators: There have been lots of articles about VO₂ actuators published since 2010, but most of them are based on bilayer or multilayered structures. According to the first two papers of Rúa et al. and Cao et al. reported the VO₂/metal actuators (J. Appl. Phys., 2010, 107, 074506; J. Appl. Phys., 2010, **108**, 083538), the performance of these bimorph actuators is highly associated with the thickness ratio of the two materials and the quality of their interface. The interfacial problems greatly limit the lifetime of VO₂ actuators. For example, the VO₂/Cr interface tends to be broken because of the oxidation of VO₂ that is accelerated in environments with high humidity. As shown in Figure R14, the supporting Cr layer is peeled off from the device due to the damaged VO₂/Cr interface after a certain number of oscillation cycles. In addition, most bimorph actuators are initially curved due to a remarkable residual stress in the VO₂/Cr interface at room temperature, which brings a considerable difficulty in controllable design of VO₂ bimorph micro-robotics (Appl. Phys. Lett. 2016, **109**, 023504). In addition, the curved device is always in a high-energy strained state that could accelerate the fatigue of the VO₂/Cr interface and decrease the lifetime of device.

Figure R14. SEM image of an aged VO₂/Cr bimorph actuator with a shedding Cr layer.

To address the above problems in VO₂ bimorph actuators, we proposed a totally new kind of actuators composed of a single VO₂ beam utilizing the lateral oxygen gradient, so-called SCVA in our recent work (Adv. Funct. Mater., 2019, **29**, 1900527). This actuator has a simple single-crystalline structure and thus has a better stability than the bimorph devices. Most importantly, the SCVAs demonstrate almost the same or superior actuation performance to the optimized VO₂/Cr actuators. These SCVAs have many other merits such

as, ultra-high response speed, straight shape at initial and final stages of actuation, predictable deflection amplitude, etc.

It is noted that the actuation is completely driven by the asymmetric evolution of sub-micrometer-scale M2, M1, T, and R domains upon the change in temperature. We are not sure whether this actuation mechanism has been similarly achieved in other actuator systems, but we are quite sure that it is a new idea for develop controllable and high performance VO₂ actuators.

b) SCVAs achieve the best theoretical performance: Most of the reported VO₂ actuators, e.g., the reported M1-R SCVA, can only utilize 1% strain of M1-R transition. In 2013, Wang et al. (ACS Nano, 2013, 7, 2266–2272) reported that the M2-R transition with a unidirectional strain of ~2% greatly promoted the performance of VO₂/Cr bimorph actuators. However, the M2 phase in the bimorph actuators only transiently perform under the large interfacial stress, so that the complicated actuation behavior of this M2-R transition-driven device is uncontrollable and unpredictable. After this, the VO₂ actuators driven by the M2-R transition have been seldomly reported, which may be attributed to the tough requirements of stress for the stabilization of metastable M2 phases. Notably, in this work, we provide a reliable way to fabricate the M2-R transition-driven SCVAs. As expected, the M2-R SCVA demonstrates almost the best actuation performance among the reported VO₂ actuators as shown in Figure R15. Compared with the mentioned bimorph devices, the bidirectional actuation mode of M2-R SCVAs is consistent with other SCVAs and their actuation performance can be directly modulated by the reaction conditions, enabling their functionalized applications in practical devices.

Figure R15. Actuation frequency and volumetric work density for various actuator systems, including ferroelectric/piezoelectric (FE/PE) oxides, polymers, shape memory alloys (SMAs), VO₂ bimorph actuators, and SCVAs.

c) **Diversified functions of SCVA family:** This work introduces three members of the SCVA family, M1-R SCVA, T-M2 SCVA, and M2-R SCVA, that utilize different phase transitions to drive the bidirectional bending actuation of VO₂ beams. Despite similar actuation modes and working temperature windows, they demonstrate different characteristics. Among them, the M2-R SCVA demonstrates the best actuation performance, as introduced before, and provides the highest energy conversion efficiency of ~2.43%, enabling the largest dislocation and the highest output work. As for the T-M2 SCVA, it cannot provide a large output, but it consumes far less energy than the other SCVAs. Therefore, the T-M2 SCVA can be used as an energy-saving actuation device and its working temperature range can be easily modulated by adjusting the reaction conditions. In contrast, the M1-R SCVA does not have a specific advantage in performance, but we should admit that it is the easiest one to fabricate due to the best stability of M1 and R structures. It is desired to include new VO₂ phases or develop new assembly way of the phase domains to enrich the functions of SCVA family.

Based on the above points, we conclude that this work, “Phase Management in Single-Crystalline Vanadium Dioxide Beams”, introduces a new degree of freedom in 1D VO₂ structures by stabilizing and engineering multiple phase domains in as-grown non-stoichiometric VO₂ beams. The enriched phase family supplies us with various functional units at the sub-micrometer level for the extended study on the applications of VO₂. It is believed that the intentional design of the phase domain behavior in single-crystalline VO₂ beams through the present stoichiometry engineering strategy can be used to discover new physical properties and promising applications of VO₂. We also anticipate this work paves a new way to an improved understanding of strongly correlated systems in general.

We respect and understand different opinions from the reviewers, which did help us think more and deeper about our results. We expect that the above discussion could help reviewers to gain a further picture on the novelty of this work and its significance to the phase transition modulation/applications of VO₂ materials. We hope that the editors and the reviewers can find the revised manuscript suitable for further consideration by Nature Communications.

We look forward to receiving any further comments and positive feedback.

Sincerely,

Chun CHENG (Ph.D.)

Associate Professor
Department of Material Sciences and Engineering
Southern University of Science and Technology
P.R. China

REVIEWERS' COMMENTS

Reviewer #1 (Remarks to the Author):

I am happy with the revisions and the responses to my comments and I suggest publish this manuscript on Nature Communications.

Reviewer #2 (Remarks to the Author):

The authors have extensively revised the manuscript and my concerns are all addressed. I believe this work is now ready for publication at Nature Commun.

Reviewer #3 (Remarks to the Author):

The authors have carefully considered my earlier concerns about the lack of novelty in the manuscript and have extensively revised the manuscript to reflect the importance and novelty of their work.

I also read through the comments from other reviewers and the responses from the authors. I believe the authors have satisfactorily responded to the comments and concerns and the manuscript is vastly improved from the previous version.

I recommend publication of the current version.